# ON THE ELIGIBILITY OF LLMS FOR COUNTERFACTUAL REASONING: A DECOMPOSITIONAL STUDY

**Shuai Yang**[1]   **Qi Yang**[2]   **Luoxi Tang**[1]   **Yuqiao Meng**[1]   **Nancy Guo**[1]
**Jeremy Blackburn**[1]   **Zhaohan Xi**[1]

[1]Binghamton University   [2]Shanghai University
 `zxi1@binghamton.edu`

## ABSTRACT

Counterfactual reasoning has emerged as a crucial technique for generalizing the reasoning capabilities of large language models (LLMs). By generating and analyzing counterfactual scenarios, researchers can assess the adaptability and reliability of model decision-making. Although prior work has shown that LLMs often struggle with counterfactual reasoning, it remains unclear which factors most significantly impede their performance across different tasks and modalities. In this paper, we propose a decompositional strategy that breaks down the counterfactual generation from causality construction to the reasoning over counterfactual interventions. To support decompositional analysis, we investigate 11 datasets spanning diverse tasks, including natural language understanding, mathematics, programming, and vision-language tasks. Through extensive evaluations, we characterize LLM behavior across each decompositional stage and identify how modality type and intermediate reasoning influence performance. By establishing a structured framework for analyzing counterfactual reasoning, this work contributes to the development of more reliable LLM-based reasoning systems and informs future elicitation strategies.

## 1 INTRODUCTION

Large language models (LLMs) have exhibited remarkable proficiency across a diverse range of tasks, including natural language understanding (Devlin et al., 2019; Kuang et al., 2025), multimodal reasoning (Hu et al., 2017; Lu et al., 2022; Yang et al., 2023; Zhou et al., 2024; Tang et al., 2025b), and domain-specific generation (Desai et al., 2026; Kalvakuntla et al., 2026; Tang et al., 2025a). Despite these advancements, concerns persist regarding their reasoning and generalization capabilities. A particularly challenging aspect of model evaluation is **Counterfactual Reasoning**, i.e., the ability to adjust responses when presented with modified premises (Pearl & Mackenzie, 2018) (e.g., *What is the outcome in a hypothetical condition?*). Investigating the counterfactual reasoning of LLMs provides an interpretable step to understand their adaptability under hypothetical alterations to input conditions (Gat et al., 2024; Huang et al., 2024).

Prior studies have demonstrated that LLMs often struggle with counterfactual reasoning and frequently fail to maintain logical consistency or adjust to context shifts (Li et al., 2023; Nguyen et al., 2024; Wang et al., 2024). While these works highlight notable performance gaps, they lack a standardized framework for systematically analyzing and understanding counterfactual behaviors in LLMs. Consequently, it remains unclear what factors most significantly impact LLM performance in counterfactual scenarios. Furthermore, counterfactual reasoning has often been evaluated in a direct and monolithic manner, primarily by introducing interventions and assessing model responses (Li et al., 2023), without grounding the analysis in the underlying causal structure that gives rise to such interventions. This overlooks the foundational role of causal modeling. Specifically, the identification of causal variables and their dependencies are essential for understanding counterfactuals.

To address these gaps, we are motivated by the structural formulation of counterfactual reasoning under the Structural Causal Model (SCM) formalism (Pearl, 2009), in which counterfactual reasoning

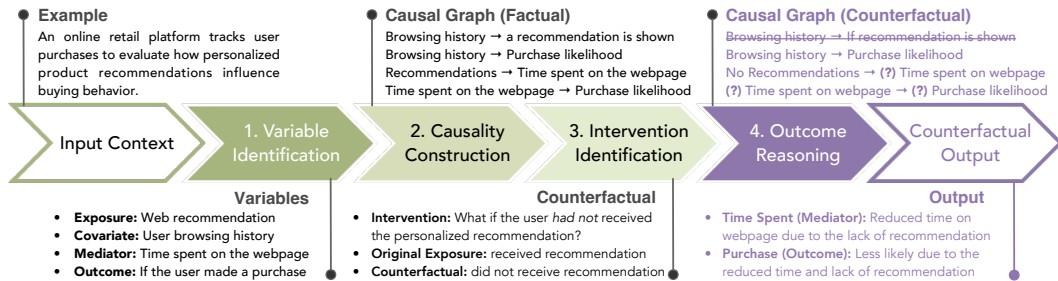

Figure 1: A workflow and illustrative example that decomposes LLM-based counterfactual reasoning into four stages: (1) identifying causal variables (e.g., *whether web recommendation is shown*), (2) constructing the causal graph (e.g., *browsing history → a recommendation is shown*), (3) specifying the counterfactual intervention (e.g., *no recommendation shown*), and (4) reasoning about the counterfactual outcome (e.g., *less likely to purchase a product online*).

must proceed through a sequence of regularized steps including inferring latent variables from observations, modifying the SCM via intervention, and computing the updated outcome. Accordingly, we outline a **Decompositional Strategy** that breaks down the analysis of counterfactual reasoning into distinct stages. Our approach departs from prior work that focuses solely on counterfactual generation. Instead, we begin by examining the causal structure of factual conditions, which serves as the necessary foundation for valid counterfactual reasoning.

As illustrated in Figure 1, our methodology is outlined into four stages. First, we assess (i) whether LLMs can accurately identify the four variable groups critical to causal reasoning: Exposure, Covariate, Mediator, and Outcome. Next, we evaluate (ii) whether LLMs can correctly construct a corresponding causal graph in the form of a directed acyclic graph (DAG). Building on this causality modeling, we then study LLMs' counterfactual reasoning abilities by evaluating (iii) whether they can identify the correct intervened variable (i.e., the Exposure), and (iv) whether they can accurately infer the counterfactual mediators and final outcomes by reasoning over the updated causal graph.

To support our decompositional study, we construct a benchmark by collecting and curating 11 counterfactual datasets across diverse tasks, including natural language understanding, mathematics, programming, and vision-language reasoning. We curate each dataset by extracting factual and counterfactual variables, identifying causal elements, and constructing corresponding causal graphs as reference structures for evaluation purpose. In experiments, we test the performance of leading LLMs across each decompositional stage to analyze their sufficiency in handling individual reasoning components. Based on the observed performance among these decomposed evaluations, we propose targeted improvements, such as integrating modality-specific function-calling interfaces within a tool-augmented learning paradigm, to address critical reasoning bottlenecks. Additionally, we evaluate the impact of different elicitation (prompting) strategies, including Chain-of-Thought (CoT) (Wei et al., 2022), Chain-of-Thought with Self-Consistency (CoT-SC) (Wang et al., 2022), and Tree-of-Thought (ToT) (Yao et al., 2023) reasoning. Collectively, our evaluations provide a solid step for understanding and enhancing LLMs in complex reasoning tasks and imaginative scenarios.

In summary, we make the following contributions:

- **Decompositional Framework**–We propose a decompositional strategy that spans from causal modeling to counterfactual reasoning, enabling a systematic evaluation of LLMs' capabilities in understanding and performing counterfactual tasks.

- **Benchmark Construction**–We construct a comprehensive evaluation benchmark by curating causal structures and counterfactual instances across multiple domains. This benchmark standardizes decompositional evaluations and supports consistent analysis across tasks and modalities.

- **Evaluation and Improvement Strategy**–We evaluate leading LLMs under diverse tasks. By identifying LLMs' capabilities in specific decompositional stage, we propose actionable strategies to improve LLMs' counterfactual adaptability.

## 2 RELATED WORK

**Counterfactual Reasoning.** A fundamental component of causal inference is to examine hypothetical scenarios, which addresses the question: *What would have occurred had a particular factor or decision differed?* (Pearl & Mackenzie, 2018) This method facilitates causal analysis by comparing observed outcomes with those projected under alternative conditions. Empirical research has demonstrated the broad applicability of counterfactual reasoning across multiple domains, including healthcare, business, and fairness (Gvozdenović et al., 2021; Kyrimi et al., 2025; Gow et al., 2016; Kasirzadeh & Smart, 2021; Koonce et al., 2011). This stands in contrast to many contemporary AI systems, which predominantly rely on statistical correlations while lacking robust capacities for abstract reasoning and causal inference (Jiao et al., 2024).

**Counterfactual Reasoning in AI and NLP.** Counterfactual reasoning has emerged as a powerful framework for enhancing model interpretability and causal understanding in AI and NLP. In medical AI, SyncTwin (Qian et al., 2021) proposed a counterfactual estimation framework that constructs synthetic patient data to predict potential outcomes under alternative treatments. In NLP, a Counterfactual Reasoning Model (CRM) (Feng et al., 2021) is developed using LLMs to generate contrastive samples, improving sentiment analysis and inference tasks. There are also order-faithfulness metrics (Gat et al., 2024) to evaluate causal explanations in black-box models. These contributions demonstrate the versatility of counterfactual methods in improving model transparency and reliability across domains.

**LLMs and Elicitation.** Recent advances have significantly enhanced LLMs' reasoning capabilities through several elicitation (prompting) approaches. The introduction of Chain-of-Thought (CoT) prompting (Wei et al., 2022) and its extensions, the Self-Consistency CoT (Wang et al., 2022) and Tree-of-Thought (ToT) (Yao et al., 2023), have enabled more structured and reliable multi-step reasoning. These innovations collectively represent a paradigm shift from simple pattern recognition to deliberate, verifiable reasoning in LLMs (Meng et al., 2025; Liu et al., 2025).

**Evaluation of Counterfactual Reasoning.** Evaluating counterfactual reasoning in LLMs has garnered growing attention. However, most prior works assess counterfactual through end-to-end evaluations such as contrastive counterfactuals (e.g., "What would happen if X didn't occur?") (Huang et al., 2024; Frohberg & Binder, 2021; Zhang et al., 2024b; Le et al., 2023). Other studies such as MalAlgoQA (Sonkar et al., 2024) introduce the concept of algorithms to assess LLMs' ability to reason about flawed hypothetical paths. Their setup focuses on identifying distractor rationales in multiple-choice formats, revealing LLM struggles in understanding student misconceptions Other efforts like DICE (Shrivastava & Aoyagui, 2025) and CausalProbe (Chi et al., 2024) create diagnostic benchmarks to evaluate causal sensitivity or counterfactual faithfulness in static question-answering formats, without decomposing the reasoning process into interpretable modules . Similarly, synthetic datasets have been used to analyze RNN inductive biases in agreement prediction (Ravfogel et al., 2019) but do not extend to structured counterfactual inference tasks. Compared to these works, our study introduces a modular evaluation aligned with Pearl's structural causal model (Pearl & Mackenzie, 2018). We decompose counterfactual reasoning into four interconnected sub-tasks to open up fine-grained attribution of model failures and provide a more diagnostic and interpretable assessment of LLM reasoning capabilities.

## 3 METHODOLOGY: DECOMPOSING COUNTERFACTUAL REASONING

This section presents our methodology for decomposing counterfactual reasoning. We begin by introducing foundational concepts in causality and counterfactual reasoning (Section 3.1). Subsequently, we detail the evaluation tasks used to assess models (Section 3.2) and our corresponding construction of benchmarks over multimodal datasets (Section 3.3).

### 3.1 PRELIMINARY: FROM CAUSALITY TO COUNTERFACTUAL REASONING

**Causality.** Causality depicts the dependencies about how one variable influences another, i.e., the underlying causal effects. There are four types of variables commonly used in causal analysis: exposure, covariate, mediator, and outcome. Specifically: *(1) Exposure* (or treatment, intervention, denoted $X$) refers to the action or condition imposed on a system; *(2) Outcome* ($Y$) denotes the resulting response or effect influenced by the exposure; *(3) Covariate* ($Z$) is the pre-treatment variable

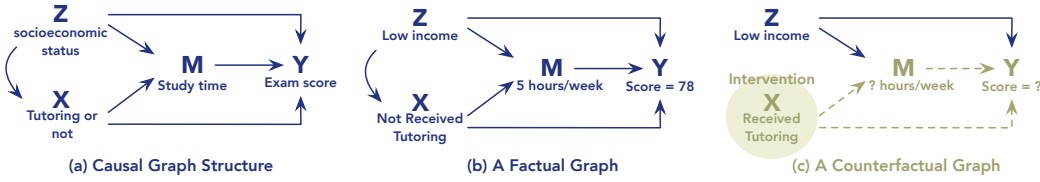

Figure 2: (a) Causal graph structure and (b)(c) A factual/counterfactual example.

that may influence both $X$ and $Y$; *(4) Mediator* ($M$) lies on the causal pathway from $X$ to $Y$, representing intermediate mechanisms through which the exposure exerts its influence.

> **Example 1** Consider a dataset that records students' academic performance in the presence of a tutoring tool. Here, the exposure $X$ indicates whether a student used the tool. The outcome $Y$ corresponds to the student's final exam score. The covariate $Z$ may include socioeconomic factors (e.g., parental income), which could influence both the use of tutoring and academic outcomes. The mediator $M$ is the number of hours the student spends studying per week.

**Causal Graph.** The relationships among exposure, covariate(s), mediator(s), and outcome(s) can be formally represented using a directed acyclic graph (DAG), commonly named a *causal graph* (Pearl & Mackenzie, 2018) that captures causal relationships, where the exposure $X$ influences the outcome $Y$ both directly and indirectly through a mediator $M$. Covariate $Z$ may affect $X$, $M$, and $Y$, as illustrated in Figure 2.

> **Example 2** The corresponding causal graph, illustrated in Figure 2-(a), would include: an arrow from $X \rightarrow M$ (e.g., tutoring influences study time), $M \rightarrow Y$ (study time influences exam performance), $X \rightarrow Y$ (direct effect of tutoring on scores), $Z \rightarrow X$ and $Z \rightarrow Y$ (e.g., socioeconomic status affects both tutoring usage and academic performance).

**Counterfactual Reasoning.** Counterfactual reasoning aims to answer:

> **Given an observed instance $(X = x, Z = z, M = m, Y = y)$, what would the outcome $Y$ be if the exposure $X$ were set to a different value $x'$, while keeping the covariate $Z$ fixed?**

The observed instance $(x, z, m, y)$ is also known as the **factual** case. In contrast, counterfactual reasoning seeks to determine the outcomes under an alternate intervention, that is, when $X = x'$.

In causal graph, we assume a two-stage causal mechanism: (1) Mediator Function: $M = f_M(X, Z)$ and (2) Outcome Function: $Y = f_Y(X, M, Z)$. Then, the counterfactual outcome under an alternative exposure $x'$ can be computed via: $Y_{x'} = f_Y(x', f_M(x', z), z)$, where we simulate a new mediator value $M_{x'} = f_M(x', z)$ based on the counterfactual exposure $x'$ and observed covariate $z$. Then, we predict the counterfactual outcome $Y_{x'}$ using the counterfactual exposure $x'$, the simulated mediator $M_{x'}$, and the same covariate $z$.

> **Example 3** As exemplified in Figure 2-(b), a student with $z = $ LOW-INCOME socioeconomic status did not receive tutoring ($x = 0$), studied 5 hours per week ($m = 5$), and scored 78 on the exam ($y = 78$).
> **We now ask:** *What would the student's score have been if they had received tutoring ($x' = 1$)?*
> **We compute:** (i) Simulated study time: $m' = f_M(x' = 1, z = $ LOW-INCOME$) = 9$ and (ii) counterfactual score: $Y_{x'=1} = f_Y(x' = 1, m' = 9, z = $ LOW-INCOME$) = 85$.
> **We conclude:** *The tutoring would have increased the student's score from 78 to 85 (Figure 2-(c)).*

### 3.2 DECOMPOSITIONAL EVALUATION TASK

Counterfactual reasoning is often described as a structured chain of analysis from identifying variables to modeling causal relations, specifying interventions, and simulating outcomes (Pearl, 2009; Bareinboim et al., 2022). Recent work in ML and LLMs further emphasizes the importance of

Table 1: Summary of counterfactual benchmarks including data source, use case, presence of causal variables (●: present, ◐: partially present), the definition of counterfactual condition, included modalities, and number of instances. Concrete examples are shown in Appendix A.

| Data | Use Case | Causal Variable | | | | Counterfactual Condition | Modality | Num |
|------|----------|---|---|---|---|--------------------------|----------|-----|
| | | $X$ | $Z$ | $M$ | $Y$ | | | |
| CRASS (Frohberg & Binder, 2021) | Question answering | ● | ◐ | ◐ | ● | "What if ..." condition | Text | 274 |
| CLOMO (Huang et al., 2024) | Text logic parsing | ● | ● | ● | ● | New premise for textual statement | Text | 1,100 |
| RNN-Typology (Ravfogel et al., 2019) | Text syntax parsing | ● | ● | ● | ● | New syntactic structure of sentence | Text | 584 |
| CVQA-Bool (Zhang et al., 2024b) | Question answering | ● | ◐ | ◐ | ● | Hypothetical behavioral pattern | Text,Image | 1,130 |
| CVQA-Count (Zhang et al., 2024b) | Numerical reasoning | ● | ◐ | ◐ | ● | Hypothetical numerical pattern | Text,Image | 2,011 |
| COCO (Le et al., 2023) | Text-image matching | ● | ● | ◐ | ● | "What if ..." condition | Text,Image | 17,410 |
| Arithmetic (Wu et al., 2024) | Mathematical reasoning | ● | ● | ● | ● | Change number base | Symbol | 6,000 |
| MalAlgoQA (Sonkar et al., 2024) | Question Answering | ● | ◐ | ◐ | ● | "What if ..." condition | Text,Symbol | 807 |
| HumanEval-Exe (Chen et al., 2021) | Code Execution simulation | ● | ◐ | ● | ● | Hypothetical coding criterion | Text,Code | 981 |
| Open-Critic (Vezora, 2024) | Code generation | ● | ◐ | ● | ● | Hypothetical descriptive functions | Text,Code | 8,910 |
| Code-Preference (Vezora, 2024) | Code summarization | ● | ◐ | ◐ | ● | Hypothetical code structures | Text,Code | 9,389 |

disentangling these steps to evaluate reasoning capacity (Kiciman et al., 2023; Chi et al., 2024). Motivated by the need for decomposition, we design four evaluation tasks that reflect the full pipeline of counterfactual reasoning, with each task targeting a distinct capability in counterfactual analysis.

- **Task I: Causal Variable Identification.** Given inputs containing factual information, a model is required to identify the values of the causal variables $(X, Z, M, Y)$. This step serves as the foundation for subsequent causal modeling and counterfactual reasoning.

- **Task II: Causal Graph Construction.** Given the identified variables, the model is tasked with constructing a DAG that captures the causal relationships among them. This step evaluates the model's ability to discover causal dependencies.

- **Task III: Counterfactual Identification.** Given a counterfactual query (e.g., "*What if variable X had been different?*"), the LLM must identify the new value of (i.e., the intervention). This task evaluates whether the model can detect intervention in the counterfactual condition.

- **Task IV: Outcome Reasoning.** Based on the constructed causal graph and identified intervention, the model is prompted to predict the counterfactual outcome. This step measures whether the model can simulate the hypothetical scenario while respecting the underlying causal mechanisms.

## 3.3 BENCHMARKING COUNTERFACTUALS

Next, we introduce the datasets we leverage for the decompositional evaluations:

**Data Sources and Use Cases.** As shown in Table 1, we collect a diverse set of datasets to ensure broad coverage across various NLP tasks and modalities. The included use cases are: **(1) Question Answering**, evaluated using CRASS (Frohberg & Binder, 2021), CVQA-Bool (Zhang et al., 2024b), and MalAlgoQA (Sonkar et al., 2024), which involve answering general-purpose textual or visually grounded questions; **(2) Text Parsing**, using CLOMO (Huang et al., 2024) for logical structure reconstruction and RNN-Typolog (Ravfogel et al., 2019) for syntactic structure understanding; **(3) Reasoning Tasks**, with CVQA-Count (Zhang et al., 2024b) for numerical reasoning and Arithmetic (Wu et al., 2024) for symbolic arithmetic computation; **(4) Multimodal Matching**, represented by the COCO dataset (Le et al., 2023) for image-text alignment; **(5) Code-based Tasks**, including HumanEval-Exe (Chen et al., 2021) for execution simulation, Open-Critic (Vezora, 2024) for generation, and Code-Preference (Vezora, 2024) for summarization. These datasets are therefore intentionally positioned as a prerequisite stage to support safe and informed downstream application.

These datasets span four modalities, natural language text, images, mathematical symbols, and code, that encompass diverse definitions of counterfactual interventions tailored to each task. Collectively, they support a comprehensive multimodal evaluation of LLMs' abilities to reason under varied counterfactual settings and data types.

**Our Preprocessing.** To support our decompositional evaluations, we curate those datasets to augment each instance with three additional aspects of information relevant to Tasks I–III (Section 3.2). Specifically, we begin by identifying and annotating the causal variables $(X, Z, M, Y)$ from the original data, questions, or descriptions. Using these annotations, we construct a DAG to represent

Table 2: LLMs' performance in causal variable identification, we report means of F1 across all instances for each variable. Each value is scaled to 100%. The standard deviation is in Table 8.

| Dataset | GPT-5 | | GPT-o4 | | Qwen3 | | Llama4-S | | Llama4-M | | Gemini2.5 | | DeepSeek | |
|---|---|---|---|---|---|---|---|---|---|---|---|---|---|---|
| | $v_1$ | $v_2$ | $v_1$ | $v_2$ | $v_1$ | $v_2$ | $v_1$ | $v_2$ | $v_1$ | $v_2$ | $v_1$ | $v_2$ | $v_1$ | $v_2$ |
| $v_1 = X$ (Exposure), $v_2 = Z$ (Covariate) | | | | | | | | | | | | | | |
| CRASS | 92.3 | 91.1 | 91.0 | 89.2 | 87.3 | 85.4 | 88.5 | 86.9 | 90.6 | 89.1 | 88.6 | 86.2 | 89.5 | 87.1 |
| CLOMO | 89.8 | 87.6 | 88.1 | 85.6 | 83.5 | 81.9 | 87.1 | 85.3 | 88.8 | 87.0 | 84.3 | 82.8 | 86.4 | 84.2 |
| RNN-Topo | 87.9 | 85.4 | 85.9 | 83.7 | 80.4 | 78.6 | 84.3 | 82.6 | 85.7 | 84.2 | 81.7 | 80.1 | 83.8 | 82.0 |
| CVQA-Bool | 79.4 | 76.2 | 79.8 | 76.5 | 72.3 | 70.5 | 77.9 | 75.8 | 79.1 | 76.9 | 68.5 | 66.9 | 70.7 | 68.3 |
| CVQA-Count | 74.7 | 72.3 | 74.3 | 72.9 | 68.9 | 67.2 | 73.6 | 71.9 | 74.4 | 72.5 | 65.8 | 63.7 | 67.4 | 65.1 |
| COCO | 72.8 | 70.2 | 73.2 | 71.1 | 67.2 | 65.4 | 72.5 | 70.8 | 73.6 | 71.7 | 62.6 | 60.9 | 65.9 | 63.2 |
| Arithmetic | 88.2 | 86.5 | 84.9 | 82.8 | 75.7 | 73.8 | 80.3 | 78.6 | 81.6 | 79.8 | 76.9 | 74.5 | 78.3 | 76.1 |
| MalAlgoQA | 84.1 | 81.3 | 81.5 | 78.9 | 72.9 | 70.6 | 79.5 | 77.1 | 80.6 | 78.2 | 73.5 | 71.2 | 75.9 | 73.4 |
| HumanEval-Exe | 69.3 | 66.9 | 71.4 | 69.2 | 63.7 | 61.9 | 67.8 | 65.7 | 68.9 | 66.7 | 59.6 | 57.3 | 62.1 | 59.8 |
| Open-Critic | 71.7 | 69.4 | 70.1 | 67.3 | 61.8 | 59.7 | 66.5 | 64.7 | 67.6 | 65.8 | 57.3 | 55.9 | 60.4 | 58.1 |
| Code-Preference | 49.6 | 68.4 | 80.2 | 69.0 | 72.9 | 60.5 | 73.6 | 61.9 | 75.2 | 63.4 | 68.4 | 66.5 | 61.2 | 59.3 |
| $v_1 = M$ (Mediator), $v_2 = Y$ (Outcome) | | | | | | | | | | | | | | |
| CRASS | 87.4 | 91.7 | 84.1 | 89.3 | 72.8 | 79.9 | 81.2 | 87.4 | 82.4 | 88.6 | 74.1 | 81.6 | 76.2 | 83.5 |
| CLOMO | 83.1 | 89.4 | 81.3 | 87.9 | 68.9 | 77.4 | 77.2 | 84.9 | 78.6 | 86.0 | 71.2 | 78.9 | 73.5 | 80.7 |
| RNN-Topo | 81.7 | 86.3 | 79.4 | 85.6 | 67.1 | 75.8 | 75.3 | 83.1 | 76.6 | 84.4 | 69.3 | 76.4 | 71.5 | 78.9 |
| CVQA-Bool | 73.5 | 79.3 | 73.9 | 80.1 | 59.7 | 66.9 | 71.8 | 78.3 | 72.9 | 79.4 | 57.3 | 63.2 | 60.5 | 68.4 |
| CVQA-Count | 69.6 | 75.2 | 70.1 | 76.2 | 56.8 | 63.6 | 68.9 | 74.8 | 70.2 | 75.9 | 54.7 | 60.4 | 57.9 | 65.1 |
| COCO | 67.3 | 73.4 | 68.0 | 74.4 | 54.2 | 61.8 | 66.2 | 72.1 | 67.4 | 73.5 | 52.8 | 58.3 | 55.7 | 62.6 |
| Arithmetic | 82.1 | 85.6 | 82.3 | 73.4 | 63.6 | 71.9 | 79.1 | 85.1 | 80.3 | 86.0 | 62.8 | 73.5 | 65.7 | 74.9 |
| MalAlgoQA | 79.2 | 83.4 | 76.8 | 80.4 | 61.5 | 69.3 | 76.6 | 81.2 | 77.8 | 82.4 | 60.2 | 70.5 | 63.8 | 72.3 |
| HumanEval-Exe | 63.2 | 67.4 | 66.0 | 70.3 | 51.9 | 59.7 | 61.9 | 66.3 | 63.0 | 67.5 | 49.7 | 57.0 | 53.6 | 60.5 |
| Open-Critic | 66.3 | 70.6 | 64.1 | 68.9 | 49.7 | 57.2 | 64.7 | 69.0 | 65.6 | 70.0 | 47.3 | 54.8 | 51.3 | 58.7 |
| Code-Preference | 65.9 | 75.3 | 66.3 | 77.0 | 50.4 | 78.6 | 64.5 | 79.3 | 65.7 | 80.4 | 48.2 | 76.1 | 52.5 | 59.8 |

the underlying causal structure of each data instance, which enrich original instances with causal and counterfactual structures. A running example in Figure 2.

**Preprocessing Feasibility.** Notably, all datasets are built upon "what-if" conditions or hypothetical scenarios (as outlined in Table 1) and intervention-style narratives, thus naturally supporting counterfactual interventions. For each instance, we parse and extract the intervened variables and, guided by the previously constructed DAG, annotate the corresponding counterfactual outcomes and construct a matched counterfactual graph. We provide the curated instances in Appendix A.

## 4 EXPERIMENT

We aim to empirically answer two research questions: **RQ$_1$:** How well do LLMs perform when their counterfactual reasoning is decomposed into distinct reasoning tasks? **RQ$_2$:** What auxiliary techniques can improve LLMs' counterfactual reasoning? We defer the experimental settings into Appendix B.1 and additional results into Appendix B.2.

**LLMs.** We evaluate reasoning-centric and multimodal LLMs due to the nature of counterfactual reasoning tasks. Specifically, we leverage GPT-5, GPT-o4-mini-high, Qwen3-VL-235B-A22B-Thinking, Llama-4-Scout-17B, Llama-4-Maverick-17B-128E, Gemini2.5-Pro, DeepSeek-VL.

**Metrics.** We use the *F1 score* for Tasks I, II, and IV, as they involve multiple instances (e.g., $M$, $Z$, or graph edges) that require set-level evaluation. For Task III, which typically involves a single intervention on $X$, we use *accuracy* to assess whether the LLM correctly identifies the intervened $X$.

### 4.1 LLM PERFORMANCE ON DECOMPOSITIONAL TASKS (RQ$_1$)

**Setting.** We evaluate LLMs independently on each decompositional task. For each task, we explicitly provide the ground-truth outputs from the preceding tasks to isolate and measure LLMs' capabilities specific to that task. For example, when assessing models' ability to construct causal graphs, we supply the original inputs along with the ground-truth causal variables.

**Task I: Causal Variable Identification.** Table 2 presents the performance of LLMs on identifying causal variables $(X, Z, M, Y)$. We observe that model performance is strongly influenced by the modality of the dataset. Specifically, datasets involving more complex modalities (e.g., images,

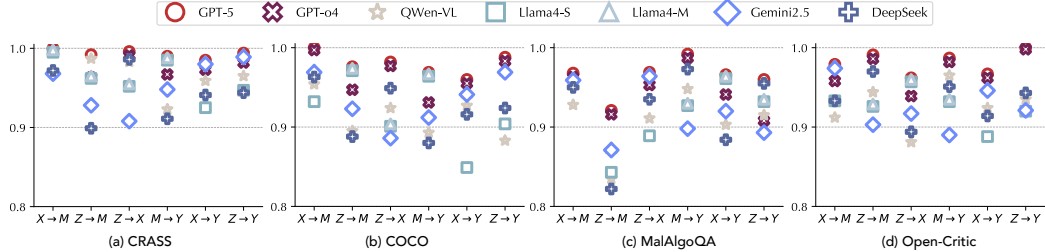

Figure 3: Evaluation on causal graph construction. We evaluate F1 score to balance (i) whether the constructed edges under one category (e.g., $X \to M$) is correctly constructed if the $(X, Z, M, Y)$ are already given. Additional results for all other datasets at Figure 5.

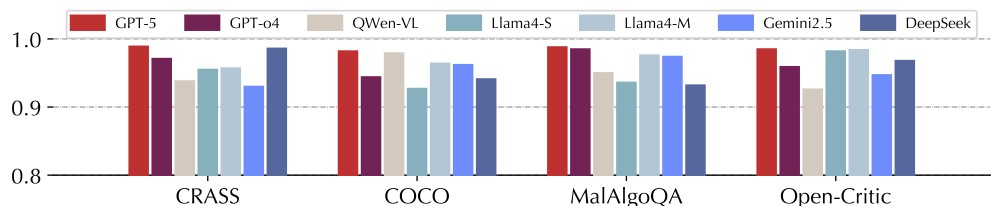

Figure 4: Evaluation of LLMs' accuracy in identifying the correct intervention (i.e., the counterfactual value of $X$). Additional results for all other datasets are provided in Figure 6.

mathematical symbols, codes) tend to reduce LLM accuracy (e.g., <0.7 F1 on Open-Critic), even when variables like $X, Z, Y$ are explicitly present in the context.

Interestingly, even within the text modality, LLMs show notable difficulty in identifying the implicit mediator $M$, which often requires reasoning about the underlying causal pathways connecting $X, Z$, and $Y$. This suggests that the challenge lies not only in the complexity of the input modality but also in the abstractness and inferential nature of the variable type itself. Together, these findings highlight the need for improved methods that enhance LLMs' capacity to handle both cross-modal complexity and deeper causal reasoning.

**Task II: Causal Graph Construction.** As described in our experimental setup, we isolate each decompositional step by providing the ground-truth outputs of preceding steps as inputs. For causal graph construction, we supply the identified variables $X, Z, M, Y$ and prompt LLMs to construct the corresponding counterfactual graph. The results are presented in Figures 3 and 5. Notably, the overall performance mostly exceeds 0.9 F1 scores, indicating that LLMs can accurately construct graph edges. Moreover, the impact of dataset modality and variable types (e.g., explicit $Z$ vs. implicit $M$) appears to be minimal in this step. We attribute this to the rule-based nature of causal graph construction: since causal graph structures are well-defined (as shown in Figure 2), it is relatively less challenging for LLMs to apply construction rules and generate the correct causal relationships.

> **Insights from causality modeling.**
>
> The major challenge in causal modeling lies in causal variable identification, where (1) LLMs are highly sensitive to the complexity and structure of the input modality, and (2) implicit variables (i.e., the mediator $M$) reveal a critical gap in LLMs' causal reasoning capabilities.

**Task III: Counterfactual Identification.** Next, we evaluate LLMs' capability in identifying interventions, i.e., determining the counterfactual values of $X$ (i.e., $X'$). As shown in Figures 4 and 6, the experimental results indicate that LLMs are generally effective at recognizing the counterfactual values of $X$ across most datasets and modalities. This demonstrates that LLMs have a solid grasp of pinpointing intervention points within the context. However, note that the task remains relatively

Table 3: LLM performance (F1 mean) in reasoning the counterfactual mediator ($M'$) and outcome ($Y'$). Standard deviation is in Table 9.

| Dataset | GPT-5 | | GPT-o4 | | Qwen3 | | Llama4-S | | Llama4-M | | Gemini2.5 | | DeepSeek | |
|---|---|---|---|---|---|---|---|---|---|---|---|---|---|---|
| | $M'$ | $Y'$ | $M'$ | $Y'$ | $M'$ | $Y'$ | $M'$ | $Y'$ | $M'$ | $Y'$ | $M'$ | $Y'$ | $M'$ | $Y'$ |
| CRASS | 92.1 | 88.0 | 90.5 | 86.2 | 80.5 | 73.9 | 70.1 | 63.5 | 84.9 | 79.5 | 81.7 | 75.2 | 82.9 | 77.1 |
| CLOMO | 90.2 | 85.3 | 88.7 | 83.9 | 77.8 | 71.6 | 67.2 | 60.9 | 82.9 | 77.2 | 79.3 | 72.8 | 80.5 | 74.3 |
| RNN-Topo | 88.9 | 83.4 | 87.9 | 81.6 | 75.6 | 69.4 | 65.2 | 58.7 | 80.5 | 75.0 | 77.1 | 70.6 | 78.3 | 72.0 |
| CVQA-Bool | 81.2 | 74.5 | 77.1 | 70.2 | 65.4 | 58.6 | 54.8 | 48.1 | 70.9 | 64.3 | 63.2 | 56.8 | 66.7 | 59.8 |
| CVQA-Count | 79.2 | 72.0 | 76.2 | 69.3 | 62.2 | 55.7 | 51.7 | 45.2 | 67.5 | 61.0 | 60.1 | 53.8 | 63.5 | 56.9 |
| COCO | 77.8 | 70.1 | 75.3 | 66.9 | 60.1 | 53.4 | 49.3 | 42.7 | 65.4 | 58.7 | 57.8 | 51.5 | 61.3 | 54.6 |
| Arithmetic | 87.8 | 82.7 | 85.8 | 80.9 | 69.8 | 63.2 | 59.1 | 52.4 | 76.3 | 70.6 | 72.1 | 65.4 | 74.0 | 67.5 |
| MalAlgoQA | 85.1 | 79.6 | 83.6 | 77.8 | 67.5 | 60.9 | 57.4 | 50.7 | 74.0 | 68.2 | 69.6 | 62.9 | 71.8 | 65.1 |
| HumanEval-Exe | 75.7 | 71.5 | 73.4 | 66.5 | 58.2 | 51.5 | 47.7 | 41.2 | 63.6 | 56.9 | 55.8 | 49.4 | 59.4 | 52.7 |
| Open-Critic | 75.3 | 69.4 | 73.8 | 67.5 | 56.0 | 49.4 | 45.8 | 39.2 | 61.5 | 54.7 | 53.7 | 47.3 | 57.2 | 50.6 |
| Code-Preference | 77.0 | 71.0 | 74.4 | 66.8 | 57.1 | 50.4 | 46.6 | 40.0 | 62.7 | 55.9 | 54.7 | 48.3 | 58.3 | 51.6 |

Table 4: Improvement in LLM performance (comparing with Table 2) for identifying explicit causal variables. Results are reported on six representative datasets spanning all major modalities.

| Dataset | GPT-o4 | | | Qwen3 | | | Llama4-S | | | Gemini2.5 | | |
|---|---|---|---|---|---|---|---|---|---|---|---|---|
| | $X$ | $Z$ | $Y$ | $X$ | $Z$ | $Y$ | $X$ | $Z$ | $Y$ | $X$ | $Z$ | $Y$ |
| CRASS | +6.0 | +6.6 | +5.5 | +10.8 | +9.4 | +9.7 | +15.2 | +11.5 | +11.9 | +6.6 | +5.3 | +7.1 |
| CLOMO | +4.7 | +5.1 | +7.9 | +12.7 | +11.3 | +12.1 | +21.0 | +14.4 | +16.8 | +6.2 | +9.4 | +15.5 |
| CVQA-Count | +17.7 | +15.9 | +18.2 | +21.9 | +21.4 | +22.7 | +32.0 | +26.1 | +24.1 | +18.1 | +15.7 | +19.3 |
| COCO | +5.5 | +3.1 | +4.4 | +8.9 | +7.6 | +8.2 | +7.2 | +6.0 | +9.7 | +6.9 | +3.8 | +5.2 |
| MalAlgoQA | +4.2 | +5.9 | +2.6 | +9.6 | +8.1 | +6.8 | +12.5 | +10.2 | +6.5 | +12.4 | +8.5 | +8.3 |
| Open-Critic | +12.8 | +14.5 | +7.0 | +15.4 | +10.9 | +8.3 | +17.4 | +5.1 | +1.2 | +7.3 | +6.1 | +9.8 |

isolated and does not challenge the model's ability to propagate the effects of the intervention through downstream variables (e.g., $M', Y'$), which we address in Task IV.

**Task IV: Outcome Reasoning.** In this final task, we evaluate LLMs' ability to infer the mediator ($M'$) and outcome ($Y'$) under a counterfactual intervention. As shown in Table 3, LLMs consistently exhibit insufficient performance in inferring these implicit variables across all datasets. Notably, since both $M'$ and $Y'$ are implicit under the counterfactual condition (whereas only $M$ is implicit in the factual condition), this result suggests that LLMs lack sufficient capacity to reason over causal chains, even when the underlying structure is explicitly provided.

> **Insights from counterfactual reasoning.**
>
> Regardless of whether the setting is factual or counterfactual, the primary challenge lies in identifying causal variables and performing causal reasoning. In particular, the complex input modality and the implicit nature of mediation hinder effective reasoning through causal pathways.

## 4.2 Auxiliary Techniques to Improve Counterfactual Reasoning (RQ$_2$)

Given the insights from previous evaluations, we aim to correspondingly address the limitations arising from multimodal complexity and intermediate reasoning, we propose augmenting LLMs with two auxiliary techniques: (i) tool-augmented execution and (ii) advanced elicitation strategies.

### 4.2.1 Tool-Augmented Execution in Explicit Variable Identification

**Settings.** To enhance LLM performance in identifying explicit variables ($X, Z, Y$) across different modalities, we adopt a tool-augmented approach, where the LLM dynamically calls additional specialized tools to assist in entity identification, identical to a named entity recognition (NER) paradigm. We leverage several pretrained models tailored to the multimodality of datasets: **(1) Text-based NER:** We use BERT-BASE-NER (Devlin et al., 2018; Tjong Kim Sang & De Meulder, 2003) to identify candidate entities in text-based data, including mathematical symbols. **(2) Vision-based NER:** We employ GROUNDING-DINO-BASE (Liu et al., 2023) to detect all relevant objects in images and generate focused regions by masking out irrelevant backgrounds. **(3) Code analysis:** We adopt

Table 5: Improvement of LLM performance (F1 score) in reasoning implicit variables.

| Dataset | Elicitation | GPT-o4 | | | Qwen3 | | | Llama4-S | | | Gemini2.5 | | |
|---|---|---|---|---|---|---|---|---|---|---|---|---|---|
| | | $M$ | $M'$ | $Y'$ | $M$ | $M'$ | $Y'$ | $M$ | $M'$ | $Y'$ | $M$ | $M'$ | $Y'$ |
| CRASS | CoT | +5.6 | +4.0 | +3.1 | +7.4 | +5.6 | +4.1 | +7.9 | +5.8 | +4.2 | +6.8 | +4.5 | +3.2 |
| | CoT-SC | +5.3 | +5.0 | +5.5 | +8.6 | +7.3 | +6.0 | +10.5 | +8.2 | +5.7 | +9.4 | +6.8 | +5.0 |
| | ToT | +6.8 | +5.2 | +4.1 | +8.9 | +6.7 | +5.0 | +8.9 | +7.4 | +4.7 | +8.2 | +5.9 | +4.3 |
| CLOMO | CoT | +6.4 | +4.9 | +3.6 | +8.3 | +6.5 | +4.8 | +8.2 | +6.9 | +5.0 | +7.6 | +5.3 | +3.9 |
| | CoT-SC | +7.0 | +5.2 | +4.2 | +9.6 | +7.6 | +5.4 | +9.9 | +7.7 | +5.5 | +8.9 | +6.5 | +4.7 |
| | ToT | +10.1 | +3.8 | +2.9 | +8.7 | +5.2 | +3.5 | +7.2 | +5.1 | +3.9 | +12.4 | +4.2 | +3.0 |
| CVQA-Count | CoT | +5.7 | +4.4 | +3.5 | +7.9 | +6.1 | +4.4 | +8.2 | +6.4 | +4.5 | +7.3 | +5.4 | +3.8 |
| | CoT-SC | +4.1 | +4.2 | +2.4 | +7.2 | +6.6 | +4.8 | +5.9 | +6.8 | +5.7 | +12.2 | +7.6 | +4.5 |
| | ToT | +4.9 | +3.8 | +3.1 | +6.9 | +5.3 | +4.0 | +6.5 | +5.2 | +4.0 | +6.1 | +4.5 | +3.1 |
| COCO | CoT | +3.8 | +2.9 | +2.1 | +6.0 | +4.6 | +3.2 | +5.4 | +4.0 | +2.6 | +4.9 | +3.4 | +2.3 |
| | CoT-SC | +5.4 | +4.1 | +3.1 | +7.2 | +5.7 | +4.3 | +7.6 | +5.9 | +3.8 | +6.9 | +5.1 | +3.6 |
| | ToT | +4.5 | +3.5 | +2.7 | +6.8 | +5.1 | +3.6 | +6.2 | +5.0 | +3.5 | +5.6 | +4.3 | +3.0 |
| MalAlgoQA | CoT | +4.6 | +3.5 | +2.7 | +7.0 | +5.5 | +4.0 | +6.6 | +5.0 | +3.3 | +5.8 | +4.0 | +2.7 |
| | CoT-SC | +5.5 | +4.3 | +3.4 | +7.7 | +6.1 | +4.6 | +7.9 | +6.3 | +4.1 | +7.2 | +5.2 | +3.7 |
| | ToT | +6.2 | +4.8 | +3.8 | +8.5 | +6.7 | +5.0 | +8.6 | +7.1 | +4.6 | +8.1 | +6.0 | +4.3 |
| Open-Critic | CoT | +3.5 | +2.7 | +2.0 | +5.3 | +4.2 | +3.0 | +5.0 | +3.7 | +2.5 | +4.5 | +3.1 | +2.1 |
| | CoT-SC | +4.2 | +3.3 | +2.6 | +6.1 | +4.9 | +3.6 | +5.8 | +4.5 | +3.1 | +5.0 | +3.5 | +2.7 |
| | ToT | +5.0 | +3.8 | +2.9 | +6.7 | +5.2 | +3.8 | +6.1 | +5.3 | +3.6 | +6.3 | +4.7 | +3.4 |

GRAPHCODEBERT (Guo et al., 2020) to extract functions, variables, and control structures for programming tasks.

After identifying candidate entities across each modality, we prompt the LLM to refine and filter the final set of explicit variables $(X, Z, Y)$ according to the formal definitions provided in Section 3.1.

**Experimental Results.** We randomly select three representative LLMs and conduct experiments across multiple datasets. As shown in Table 4, tool-augmented execution consistently improves LLM performance in identifying explicit causal variables $(X, Z, Y)$ across all modalities. For example, by leveraging GRAPHCODEBERT to parse code structures and forward the results to Llama, which gains a clearer understanding of programming logic and achieves an F1 improvement up to 0.189. Similarly, in the vision modality, the object detector GROUNDING-DINO-BASE assists by generating a set of candidate visual objects, which GPT-o4 can then contextualize and compose into a coherent factual variable. For instance, detected objects like *"Woman," "Knife,"* and *"Apple"* can be effectively integrated by GPT-o4 into the causal expression: *"A woman cuts an apple with a knife."*

These results demonstrate that tool-augmented learning effectively mitigates modality-specific bottlenecks by offloading low-level entity recognition to specialized models, allowing the LLM to focus on higher-level reasoning. Looking forward, there is potential to explore alternative tool configurations that may yield comparable or even superior performance. Additionally, future work may explore multi-agent frameworks with specialized agents to collaboratively handle different variable categories.

### 4.2.2 ADVANCED ELICITATION STRATEGIES FOR REASONING OVER IMPLICIT VARIABLES

**Settings.** To enhance LLM reasoning over implicit variables, particularly factual $M$ and counterfactual $M', Y'$, we implement advanced elicitation strategies that guide the model through more structured reasoning. Specifically, we apply Chain-of-Thought (CoT) (Wei et al., 2022), CoT with Self-Consistency (CoT-SC) (Wang et al., 2022), and Tree-of-Thought (ToT) (Yao et al., 2023):

- **CoT:** given pre-determined explicit variables, LLMs are encouraged to infer intermediate variables step-by-step.

- **CoT-SC:** LLMs are prompted to generate multiple reasoning paths and select the final answer via majority voting or consensus.

- **ToT:** LLMs are prompted to explore multiple parallel reasoning paths in a branching structure and evaluate candidate outputs based on intermediate criteria.

Table 6: Overall change of LLM performance (F1 score) with different improvement strategies. The performance change is compared with Table 3, where results of $Y'$ demonstrate the final counterfactual reasoning outcomes.

| Dataset | Improve Explicit Variable (§4.2.1) | | | | Improve Implicit Variable (§4.2.2) | | | | Improve Both | | | |
|---|---|---|---|---|---|---|---|---|---|---|---|---|
| | G5 | QW3 | LM4S | GM2.5 | G5 | QW3 | LM4S | GM2.5 | G5 | QW3 | LM4S | GM2.5 |
| CRASS | +1.8 | +2.3 | +3.5 | +2.0 | +6.2 | +7.4 | +7.9 | +6.8 | +9.0 | +10.1 | +10.5 | +9.3 |
| CLOMO | +2.1 | +3.1 | +4.2 | +2.6 | +5.7 | +6.3 | +6.1 | +6.5 | +9.2 | +10.4 | +11.0 | +9.8 |
| COCO | +1.2 | +1.5 | +1.9 | +1.4 | +4.1 | +5.1 | +5.9 | +5.1 | +5.8 | +6.4 | +6.7 | +6.2 |
| Open-Critic | +1.6 | +2.3 | +2.7 | +2.0 | +3.3 | +8.3 | +11.2 | +4.3 | +5.3 | +11.5 | +14.2 | +5.5 |

In implementation, both CoT-SC and ToT are executed with $k$=5 sampled reasoning paths. ToT further evaluates candidate outputs by scoring their textual similarity using BERTScore (Zhang et al., 2019), specifically assessing how well the intermediate results align with the original task statement.

**Experimental Results.** Our experimental results in Table 5 show that advanced elicitation strategies generally lead to improved performance in reasoning over implicit variables $(M, M', Y')$ despite factual or counterfactual cases. However, we also observe that more complex prompting strategies (e.g., CoT-SC and ToT) can sometimes perform slightly worse than simpler approaches (e.g., CoT). While these advanced methods encourage more exhaustive exploration of reasoning paths, they may also induce overthinking behavior in LLMs, leading the model to introduce unnecessary causal links or misinterpret the underlying problem structure. For instance, consider the following context *"A person is running a marathon and collapses."* with expected mediator *"Dehydration"*. While CoT and CoT-SC strategies correctly identify the mediator, ToT leads LLMs to overanalyze and identify *"lack of training"* or *"overexertion"* as the mediator. These choices, although related, are not directly supported by the input data and reflect an over-extension of the reasoning process.

The over-qualification of elicitation strategies (e.g., ToT) highlights that, while advanced prompting techniques can improve reasoning capabilities, they may also introduce complexities that divert the model from the most straightforward and contextually supported causal pathways. Therefore, it's crucial to balance the reasoning depth while maintaining alignment with the input data.

### 4.2.3 OVERALL PERFORMANCE IMPROVEMENT

In an end-to-end setting, we incorporate the prior improvements for explicit variable identification (Section 4.2.1), implicit variable reasoning (Section 4.2.2), and their combinatorial strategy to evaluate the overall change in counterfactual reasoning performance. Table 6 presents the results across representative datasets and LLMs.

We have to observations: (i) In general, improving implicit variable reasoning (i.e., for mediators and outcomes) yields more substantial gains in end-to-end performance, as these variables directly influence the final counterfactual predictions (Task IV). In contrast, improvements in explicit variable identification (e.g., for $X, Z, Y$) primarily strengthen the early stages of the pipeline, offering moderate but necessary support. (ii) Notably, the combined strategy achieves the highest overall improvement, although the gains are not strictly additive. This is due to accumulative reasoning errors across the decompositional steps, where inaccuracies in earlier predictions may propagate and compound through the pipeline. These findings highlight both the opportunities and challenges of modular improvements.

## 5 CONCLUSION

This work provides a decompositional framework for evaluating counterfactual reasoning in LLMs. We collect a set of datasets across multimodalities, and then curate them with reference causal variables, structured graphs, and counterfactual intervention. Next, we use our curated dataset to study the reasoning process into distinct stages from causal variable identification to outcome inference. We uncover the qualifications of current LLMs in counterfactual reasoning and where they fall short. Based on experimental insights, we further propose improvements to offer actionable insights for enhancing LLM reasoning, particularly in multimodality and implicit reasoning settings.

## ACKNOWLEDGEMENT

This work was supported by the National Science Foundation (NSF) under Grants 2046590, 2419831, and 2444759, as well as the SUNY Empire Innovation Program (EIP) awarded to N.L.G.

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

## A   COMPLEMENTARY INFORMATION OF CAUSALITY AND CAUSAL GRAPH

This appendix presents a concise overview of each dataset, followed by its causal structure and graphical representation, alongside a concrete example. For each dataset, we identify the four variable types—Exposure, Covariate, Mediator, and Outcome—and distinguish their roles in both factual and counterfactual scenarios, illustrating each with directed edges in the corresponding causal graph. In addition, we include a sample prompt used to generate responses on a simple text-parsing Q&A task dataset.

In evaluation, we rely on a universal prompt template as shown below, wherein only the task-specific contents are replaced for each dataset.

**Prompt Template.**

```
{
**Task Description**
You are asked to perform [Task I / Task II / Task III / Task IV] in
    decompositional counterfactual reasoning. Follow the definitions of
    causal variables and causal relations strictly.

**Input Context**
Here is the factual instance from the dataset:
[Insert factual context or multimodal description]

**Intermediate Outputs**
If applicable, use the following ground-truth results from previous tasks
    :
Exposure (X): [...]
Covariate(s) (Z): [...]
Mediator(s) (M): [...]
Outcome (Y): [...]

**Instruction for the Current Task**
```

```
Task I (Variable Identification): [description about domain specific
    meanings of causal variables X, Z, M, and Y and the identification
    task]
Task II (Graph Construction): [instruction about constructing the causal
    graph by listing all directed edges among X, Z, M, Y.
Task III (Intervention Identification): [instruction about identifying
    which variable is intervened on in the counterfactual query.]
Task IV (Outcome Reasoning): [instruction about inferring] the
    counterfactual mediator M' and outcome Y' under the specified
    intervention.

**Output Format**
Provide the answer using the following structure:
Exposure (X): [...]
Covariate(s) (Z): [...]
Mediator(s) (M): [...]
Outcome (Y): [...]
Causal Edges: [...]
Intervention: [...]
Counterfactual Mediator (M'): [...]
Counterfactual Outcome (Y'): [...]
}
```

**CRASS Example**. The **Counterfactual Reasoning Assessment for Structured Scenarios (CRASS)** dataset is designed to evaluate whether language models can reason about hypothetical alternatives to factual events. Each example in CRASS presents a factual scenario (e.g., *"A woman opens a treasure chest"*) followed by a counterfactual question (e.g., *"What would have happened if the woman had not opened the treasure chest?"*). Models are asked to select the most logically consistent outcome from multiple-choice options, such as *"The treasure chest would have remained closed"*, which is labeled as correct. The following displays a full example:

```
{
  "input": "A woman opens a treasure chest. What would have happened if
      the woman had not opened the treasure chest?",
  "target_scores": {
    "The treasure chest would have been open.": 0,
    "That is not possible.": 0,
    "The treasure chest would have remained closed.": 1,
    "I don't know.": 0
  }
}
```

**CRASS Causality & Causal Graph.**

```
{
  "factual_roles": {
    "Exposure": ["act of opening treasure chest"],
    "Covariate": ["key possession", "physical capability"],
    "Mediator": ["lock mechanism release"],
    "Outcome": ["chest opened"]
  },
  "counterfactual_roles": {
    "Exposure": ["omission of opening action"],
    "Covariate": ["key possession", "physical capability"],
    "Mediator": ["lock state preservation"],
    "Outcome": ["chest remains closed"]
  },
  "causal_graph": {
    "factual_edges": [
      ["key possession", "act of opening treasure chest"],
      ["key possession", "lock mechanism release"],
      ["key possession", "chest opened"],
      ["physical capability", "act of opening treasure chest"],
      ["physical capability", "lock mechanism release"],
```

```
            ["physical capability", "chest opened"],
            ["act of opening treasure chest", "lock mechanism release"],
            ["lock mechanism release", "chest opened"],
            ["act of opening treasure chest", "chest opened"],
        ],
        "counterfactual_edges": [
            ["key possession", "lock state preservation"],
            ["key possession", "chest remains closed"],
            ["physical capability", "lock state preservation"],
            ["physical capability", "chest remains closed"],
            ["omission of opening action", "lock state preservation"],
            ["lock state preservation", "chest remains closed"],
            ["omission of opening action", "chest remains closed"],
        ]
    }
}
```

**CLOMO Example.** The **Counterfactual Logical Modification (CLOMO)** dataset is designed to evaluate whether large language models can perform controlled, counterfactual edits to natural language arguments in a logically coherent way. Each example presents a base argument and two premises: Premise 1 has a logical sensitivity with the original argument, while Premise 2 does not. The model is instructed to modify the argument such that Premise 2 has a logical sensitivity with the original argument, while Premise 1 no longer is. For instance, given an argument attributing the rise in gasoline prices fully to government policies, the model must produce a revised version (e.g., changing "fully responsible" to "partly leads") of an argument that shifts logical sensitivity from one premise to another without introducing new claims. The following displays a full example:

```
{
  "instruction": "In the following, you will see an argument and 2
      premises, where Premise 1 provides a necessary assumption to the
      Argument. Please modify the Statements in the Argument until
      Premise 2 provides a necessary assumption to the Argument instead,
       while Premise 1 fails to provide a necessary assumption to the
      Argument. Note that no additional statement should be added. ",
  "input": "Argument: Statement1: Consumer advocate : there is no doubt
      that the government is responsible for the increased cost of
      gasoline, because the government's policies have significantly
      increased consumer demand for fuel, and as a result of increasing
      demand, the price of gasoline has risen steadily.Premise1: The
      government can bear responsibility for that which it indirectly
      causes.Premise2: Consumer demand for gasoline cannot increase
      without causing gasoline prices to increase.Please write the
      modified argument below: ",
  "output": "Statement1: Consumer advocate : there is no doubt that the
      government partly leads to the increased cost of gasoline, because
       the government's policies have significantly increased consumer
      demand for fuel, and as a result of increasing demand, the price
      of gasoline has risen steadily undoubtedly."
}
```

**CLOMO Causality & Causal Graph.**

```
{
  "factual_roles": {
    "Exposure": ["Premise 1 as necessary assumption"],
    "Covariate": [
      "Government's policy impact on demand",
      "Demand-price relationship assumption"
    ],
    "Mediator": ["Causal attribution mechanism (direct vs indirect)"],
    "Outcome": ["Full responsibility attribution to government"]
  },
  "counterfactual_roles": {
    "Exposure": ["Premise 2 as necessary assumption"],
```

```
      "Covariate": [
        "Government's policy impact on demand",
        "Demand-price relationship assumption"
      ],
      "Mediator": ["Responsibility attribution modifier (partial vs full)"]
        ,
      "Outcome": ["Partial responsibility attribution to government"]
    },
  "causal_graph": {
    "factual_edges": [
      ["Government's policy impact on demand", "Premise 1 as necessary
          assumption"],
      ["Government's policy impact on demand", "Causal attribution
          mechanism"],
      ["Government's policy impact on demand", "Full responsibility
          attribution to government"],
      ["Demand-price relationship assumption", "Premise 1 as necessary
          assumption"],
      ["Demand-price relationship assumption", "Causal attribution
          mechanism"],
      ["Demand-price relationship assumption", "Full responsibility
          attribution to government"],
      ["Premise 1 as necessary assumption", "Causal attribution mechanism
          "],
      ["Causal attribution mechanism", "Full responsibility attribution
          to government"],
      ["Premise 1 as necessary assumption", "Full responsibility
          attribution to government"]
    ],
  "counterfactual_edges": [
      ["Government's policy impact on demand", "Responsibility
          attribution modifier"],
      ["Government's policy impact on demand", "Partial responsibility
          attribution to government"],
      ["Demand-price relationship assumption", "Responsibility
          attribution modifier"],
      ["Demand-price relationship assumption", "Partial responsibility
          attribution to government"],
      ["Premise 2 as necessary assumption", "Responsibility attribution
          modifier"],
      ["Responsibility attribution modifier", "Partial responsibility
          attribution to government"],
      ["Premise 2 as necessary assumption", "Partial responsibility
          attribution to government"]
    ]
  }
}
```

**RNN-Typology Example.** This is a synthetic dataset contains sentence pairs that reflect syntactic alterations to word orders (e.g., converting English from subject-verb-object (SVO) to subject-object-verb (SOV) order). For example, the factual sentence *"Tim saw Lucas."* (SVO) is transformed to its SOV equivalent *"Tim Lucas saw."*.

```
"tim saw lucas.": "tim lucas saw."
```

**RNN-Typology Causality & Causal Graph.**

```
{
  "factual_roles": {
    "Exposure": ["subject-verb-object order"],
    "Covariate": ["syntactic rule", "Lexical items (Tim, saw, Lucas)"],
    "Mediator": ["SOV reordering operation"],
    "Outcome": ["tim saw lucas."]
  },
```

```
    "counterfactual_roles": {
      "Exposure": ["subject-object-verb order"],
      "Covariate": ["syntactic rule", "Lexical items (Tim, saw, Lucas)"],
      "Mediator": ["SVO restoration operation"],
      "Outcome": ["tim lucas saw."]
    },
    "causal_graph": {
    "factual_edges": [
      ["syntactic rule", "subject-verb-object order"],
      ["syntactic rule", "SOV reordering operation"],
      ["syntactic rule", "tim saw lucas."],
      ["Lexical items (Tim, saw, Lucas)", "subject-verb-object order"],
      ["Lexical items (Tim, saw, Lucas)", "SOV reordering operation"],
      ["Lexical items (Tim, saw, Lucas)", "tim saw lucas."],
      ["subject-verb-object order", "SOV reordering operation"],
      ["SOV reordering operation", "tim saw lucas."],
      ["subject-verb-object order", "tim saw lucas."]
    ],
    "counterfactual_edges": [
      ["syntactic rule", "SVO restoration operation"],
      ["syntactic rule", "tim lucas saw."],
      ["Lexical items (Tim, saw, Lucas)", "SVO restoration operation"],
      ["Lexical items (Tim, saw, Lucas)", "tim lucas saw."],
      ["subject-object-verb order", "SVO restoration operation"],
      ["SVO restoration operation", "tim lucas saw."],
      ["subject-object-verb order", "tim lucas saw."]
    ]
    }
}
```

**CVQA-Bool Example. Counterfactual Visual Question Answering(CVQA)** is designed to assess the ability of vision-language models to perform counterfactual reasoning over images. Each example presents a factual visual query-answer pair (e.g., *"Is there a red sandal here?" → yes*) grounded in a COCO image, along with a corresponding counterfactual query that modifies a key visual condition (e.g., *"Would there be a red sandal here if all shoes were removed?" → no*). The task requires the model to infer changes in object presence or relationships under hypothetical alterations to the scene. The dataset focuses on a boolean query type. The following displays an example:

| real image | query | answer | new query | new answer | type |
|---|---|---|---|---|---|
|  | Is there a red sandal here? | yes | Would there be a red sandal here if all shoes were removed? | no | boolean |

**CVQA-Bool Causality & Causal Graph.**

```
{
  "factual_roles": {
    "Exposure": ["presence of red sandal"],
    "Covariate": ["original shoe collection in cart", "visual recognition
        capability"],
    "Mediator": ["sandal-as-shoe categorical inclusion"],
    "Outcome": ["yes"]
  },
  "counterfactual_roles": {
    "Exposure": ["removal of all shoes"],
    "Covariate": ["original shoe collection in cart", "visual recognition
        capability"],
    "Mediator": ["sandal-shoe categorical dependency"],
    "Outcome": ["no"]
  },
  "causal_graph": {
    "factual_edges": [
```

```
        ["original shoe collection in cart", "presence of red sandal"],
        ["original shoe collection in cart", "sandal-as-shoe categorical
            inclusion"],
        ["original shoe collection in cart", "yes"],
        ["visual recognition capability", "presence of red sandal"],
        ["visual recognition capability", "sandal-as-shoe categorical
            inclusion"],
        ["visual recognition capability", "yes"],
        ["presence of red sandal", "sandal-as-shoe categorical inclusion"],
        ["sandal-as-shoe categorical inclusion", "yes"],
        ["presence of red sandal", "yes"]
    ],
    "counterfactual_edges": [
        ["original shoe collection in cart", "sandal-shoe categorical
            dependency"],
        ["original shoe collection in cart", "no"],
        ["visual recognition capability", "sandal-shoe categorical
            dependency"],
        ["visual recognition capability", "no"],
        ["removal of all shoes", "sandal-shoe categorical dependency"],
        ["sandal-shoe categorical dependency", "no"],
        ["removal of all shoes", "no"]
    ]
  }
}
```

**CVQA-Count Example. Visual Counterfactual Query Dataset (CVQA)** also evaluates whether language models can perform a direct or indirect numerical counterfactual reasoning grounded in visual inputs. Each example consists of a factual visual question (e.g., *"How many plates are there?"* → 1) paired with a corresponding counterfactual query that modifies the quantity in a clearly defined way (e.g., *"How many plates would there be if 2 more plates were added?"* → 3). The model must integrate visual perception (e.g., detecting a single white plate in an image) with numerical logic (e.g., adding 2) to produce the correct answer. The dataset focuses on a counting query type. The following displays an example:

| real image | query | answer | new query | new answer | type |
|---|---|---|---|---|---|
|  | How many plates are there | 1 | How many plates would there be if 2 more plates were added? | 3 | direct counting |

**CVQA-Count Causality & Causal Graph.**

```
{
  "factual_roles": {
    "Exposure": ["current plate presence (1 unit)"],
    "Covariate": ["original plate count (1)", "visual counting capability
        "],
    "Mediator": ["visual plate detection mechanism"],
    "Outcome": ["1"]
  },
  "counterfactual_roles": {
    "Exposure": ["addition of 2 plates"],
    "Covariate": ["original plate count (1)", "visual counting capability
        "],
    "Mediator": ["numerical addition operation"],
    "Outcome": ["3"]
  },
  "causal_graph": {
    "factual_edges": [
      ["original plate count (1)", "current plate presence (1 unit)"],
      ["original plate count (1)", "visual plate detection mechanism"],
```

```
        ["original plate count (1)", "1"],
        ["visual counting capability", "current plate presence (1 unit)"],
        ["visual counting capability", "visual plate detection mechanism"],
        ["visual counting capability", "1"],
        ["current plate presence (1 unit)", "visual plate detection
            mechanism"],
        ["visual plate detection mechanism", "1"],
        ["current plate presence (1 unit)", "1"]
    ],
    "counterfactual_edges": [
        ["original plate count (1)", "numerical addition operation"],
        ["original plate count (1)", "3"],
        ["visual counting capability", "numerical addition operation"],
        ["visual counting capability", "3"],
        ["addition of 2 plates", "numerical addition operation"],
        ["numerical addition operation", "3"],
        ["addition of 2 plates", "3"]
    ]
  }
}
```

**COCO Example. Common Objects in Context(COCO)** dataset provides automatically constructed counterfactual examples for evaluating multimodal reasoning in image-text pairs. Each instance contains two images and two near-identical captions that differ only in a key noun (e.g., *"A big burly grizzly bear is shown with grass in the background"* vs. *"A big burly grizzly bear is shown with deer in the background"*). The dataset is designed to test whether models can detect minimal semantic changes and determine whether the new image visually aligns with the counterfactual caption. The goal is to assess visual-textual consistency and a model's sensitivity to causal or identity-based alterations in structured multimodal contexts. The following displays an example:

| Factual Caption | Image 0 | Counterfactual Caption | Image 1 |
|---|---|---|---|
| A big burly grizzly bear is shown with grass in the background. |  | A big burly grizzly bear is shown with deer in the background. |  |

**COCO Causality & Causal Graph.**

```
{
  "factual_roles": {
    "Exposure": ["original image (bear with grass)"],
    "Covariate": ["bear presence", "background context"],
    "Mediator": ["grass visual detection"],
    "Outcome": ["caption with 'grass'"]
  },
  "counterfactual_roles": {
    "Exposure": ["modified image (bear with deer)"],
    "Covariate": ["bear presence", "background context"],
    "Mediator": ["deer-grass substitution mechanism"],
    "Outcome": ["caption with 'deer'"]
  },
  "causal_graph": {
    "factual_edges": [
      ["bear presence", "original image (bear with grass)"],
      ["bear presence", "grass visual detection"],
      ["bear presence", "caption with 'grass'"],
      ["background context", "original image (bear with grass)"],
      ["background context", "grass visual detection"],
      ["background context", "caption with 'grass'"],
      ["original image (bear with grass)", "grass visual detection"],
      ["grass visual detection", "caption with 'grass'"],
```

```
          ["original image (bear with grass)", "caption with 'grass'"]
      ],
      "counterfactual_edges": [
        ["bear presence", "deer-grass substitution mechanism"],
        ["bear presence", "caption with 'deer'"],
        ["background context", "deer-grass substitution mechanism"],
        ["background context", "caption with 'deer'"],
        ["modified image (bear with deer)", "deer-grass substitution
            mechanism"],
        ["deer-grass substitution mechanism", "caption with 'deer'"],
        ["modified image (bear with deer)", "caption with 'deer'"]
      ]
    }
}
```

**Arithmetic Example. Base-computation Arithmetic** dataset evaluates counterfactual numerical reasoning by testing arithmetic operations across multiple numeral systems(e.g. Base-8, 9, 10, 11, 16). Each example pairs a factual base-10 calculation with a counterfactual alternate-base computation (e.g., base-8: $14_8 + 57_8 = 73_8$, base-16: $EC_{16} + DD_{16} = 1C9_{16}$). The dataset includes inputs (num1, num2), the numeral system (e.g., "8" for octal, "16" for hexadecimal), and the base-specific result (addrst). It assesses models' ability to adapt numeral system transitions and consistency in counterfactual reasoning. The following display a base-8 computation example and a base-16 example:

```
{
  "8": {
    "num1": "14",
    "num2": "57",
    "addrst": "73"
  },
  "16": {
    "num1": "EC",
    "num2": "DD",
    "addrst": "1C9"
  }
}
```

**Base-8 Arithmetic Causality & Causal Graph.**

```
{
  "factual_roles": {
    "Exposure": ["10-based system"],
    "Covariate": ["14", "57"],
    "Mediator": ["base-10 arithmetic operation"],
    "Outcome": ["71"]
  },
  "counterfactual_roles": {
    "Exposure": ["8-based system"],
    "Covariate": ["14", "57"],
    "Mediator": [
      "base-8 to base-10 conversion",
      "base-10 sum conversion to base-8"
    ],
    "Outcome": ["73"]
  },
  "causal_graph": {
    "factual_edges": [
      ["14", "10-based system"],
      ["14", "base-10 arithmetic operation"],
      ["14", "71"],
      ["57", "10-based system"],
      ["57", "base-10 arithmetic operation"],
      ["57", "71"],
      ["10-based system", "base-10 arithmetic operation"],
```

```
      ["base-10 arithmetic operation", "71"],
      ["10-based system", "71"]
    ],
    "counterfactual_edges": [
      ["14", "base-8 to base-10 conversion"],
      ["14", "73"],
      ["57", "base-8 to base-10 conversion"],
      ["57", "73"],
      ["8-based system", "base-8 to base-10 conversion"],
      ["base-8 to base-10 conversion", "base-10 sum conversion to base-8"
          ],
      ["base-10 sum conversion to base-8", "73"],
      ["8-based system", "73"]
    ]
  }
}
```

**Base-16 Arithmetic Causality & Causal Graph.**

```
{
  "factual_roles": {
    "Exposure": ["10-based system"],
    "Covariate": ["EC", "DD"],
    "Mediator": ["N.A."],
    "Outcome": ["N.A."]
  },
  "counterfactual_roles": {
    "Exposure": ["16-based system"],
    "Covariate": ["EC", "DD"],
    "Mediator": [
      "hex-to-decimal conversion",
      "decimal-to-hex reversion"
    ],
    "Outcome": ["1C9"]
  },
  "causal_graph": {
    "factual_edges": [
      ["EC", "10-based system"],
      ["DD", "10-based system"]
    ],
    "counterfactual_edges": [
      ["EC", "hex-to-decimal conversion"],
      ["hex-to-decimal conversion", "decimal-to-hex reversion"],
      ["EC", "1C9"],
      ["DD", "hex-to-decimal conversion"],
      ["DD", "1C9"],
      ["16-based system", "hex-to-decimal conversion"],
      ["decimal-to-hex reversion", "1C9"],
      ["16-based system", "1C9"]
    ]
  }
}
```

**MalAlgoQA Example (Malformed Algorithmic Question Answering (MalAlgoQA)** dataset is designed intentionally including factual and counterfactual rationales between multiple-choice question answering to validate a language model's ability to discern sound reasoning in the presence of rationales(factual or counterfactual). Each question is presented alongside a factual rationale that supports the correct answer (e.g., *"Correctly ordered the values from greatest to least: 276, 254, 237, 235."* → C), and is paired with counterfactual rationales (e.g., *"Ordered least to greatest"*) that correspond to plausible but incorrect or altered answers (e.g., A). Each example is decomposed into factual and counterfactual role pairs, allowing researchers to assess how changes in reasoning paths (rationales) lead to different answer choices. The following display an example of raw data and its decomposed data points:

```
{
    "Question":"Which list shows the following number in order from
        highest to lowest?",
    "Answer":"C",
    "Choice_A":" 235  237  254  276 ",
    "Choice_B":" 237  276  235  254 ",
    "Choice_C":" 276  254  237  235 ",
    "Choice_D":" 276  254  235  237 ",
    "Rationale_A":"Ordered least to greatest",
    "Rationale_B":"Ordered greatest to least by ones place.",
    "Rationale_C":"Correctly ordered the values from greatest to least:
        276, 254, 237, 235.",
    "Rationale_D":"Switched last 2 numbers."
}
```

```
{
    {
        "Question":"Which list shows the following number in order from
            highest to lowest?",
        "Answer": "C",
        "Counterfactual Answer":"A",
        "Choice_A":" 235  237  254  276 ",
        "Choice_B":" 237  276  235  254 ",
        "Choice_C":" 276  254  237  235 ",
        "Choice_D":" 276  254  235  237 ",
        "Counterfactual Rationale":"Ordered least to greatest",
        "Rationale_C":"Correctly ordered the values from greatest to
            least: 276, 254, 237, 235."
    },

    {
        "Question":"Which list shows the following number in order from
            highest to lowest?",
        "Answer":"C",
        "Counterfactual Answer":"B",
        "Choice_A":" 235  237  254  276 ",
        "Choice_B":" 237  276  235  254 ",
        "Choice_C":" 276  254  237  235 ",
        "Choice_D":" 276  254  235  237 ",
        "Counterfactual Rationale":"Ordered greatest to least by ones
            place",
        "Rationale_C":"Correctly ordered the values from greatest to
            least: 276, 254, 237, 235."
    }.

    {
        "Question":"Which list shows the following number in order from
            highest to lowest?",
        "Answer":"C",
        "Counterfactual Answer":"D",
        "Choice_A":" 235  237  254  276 ",
        "Choice_B":" 237  276  235  254 ",
        "Choice_C":" 276  254  237  235 ",
        "Choice_D":" 276  254  235  237 ",
        "Rationale_C":"Correctly ordered the values from greatest to
            least: 276, 254, 237, 235.",
        "Counterfactual Rationale":"Switched last 2 numbers."
    }
}
```

Take the first decomposed sample as an example showing **MalAlgoQA Causality & Causal Graph.**

```
{
```

```
  "factual_roles": {
    "Exposure": ["Ordered from greatest to least"],
    "Covariate": ["Number set (276, 254, 237, 235)"],
    "Mediator": ["Descending comparison logic"],
    "Outcome": ["Choice C"]
  },
  "counterfactual_roles": {
    "Exposure": ["Ordered least to greatest"],
    "Covariate": ["Number set (276, 254, 237, 235)"],
    "Mediator": ["Ascending comparison logic"],
    "Outcome": ["Choice A"]
  },
  "causal_graph": {
    "factual_edges": [
      ["Number set (276, 254, 237, 235)", "Ordered from greatest to least
          "],
      ["Number set (276, 254, 237, 235)", "Descending comparison logic"],
      ["Number set (276, 254, 237, 235)", "Choice C"],
      ["Ordered from greatest to least", "Descending comparison logic"],
      ["Descending comparison logic", "Choice C"],
      ["Ordered from greatest to least", "Choice C"]
    ],
    "counterfactual_edges": [
      ["Number set (276, 254, 237, 235)", "Ordered least to greatest"],
      ["Number set (276, 254, 237, 235)", "Ascending comparison logic"],
      ["Number set (276, 254, 237, 235)", "Choice A"],
      ["Ordered least to greatest", "Ascending comparison logic"],
      ["Ascending comparison logic", "Choice A"],
      ["Ordered least to greatest", "Choice A"]
    ]
  }
}
```

**HumanEval-Exe Example** This dataset performs a programming-related task: code execution. It is designed to probe the ability of code-execution language models to perform counterfactual reasoning in the context of program behavior. Each example consists of a function definition and a test case input, and the model is asked to predict the output under a factual assumption (e.g., Python's default 0-based indexing). The example is paired with a counterfactual version of the same test case, where a hypothetical condition is introduced—such as switching to 1-based indexing. The model must then predict the corresponding counterfactual output. For instance, given a function that checks for close floating-point elements in a list, the model is expected to reason whether the list and threshold would yield a different outcome if indexing conventions were altered. The full example is shown as follows:

```
{
    "instruction": "from typing import List\n\n\ndef has_close_elements(
        numbers: List[float], threshold: float) -> bool:\n    \"\"\"
        Check if in given list of numbers, are any two numbers closer to
        each other than\n    given threshold.\n    >>> has_close_elements
        ([1.0, 2.0, 3.0], 0.5)\n    False\n    >>> has_close_elements([1.
        0, 2.8, 3.0, 4.0, 5.0, 2.0], 0.3)\n    True\n    \"\"\"",
    "input": "[1.0, 2.0, 3.9, 4.0, 5.0, 2.2], 0.3",
    "output": "True",
    "counterfactual_output": "True"
}
```

**HumanEval-Exe Causality & Causal Graph.**

```
{
  "factual_roles": {
    "Exposure": ["0-based indexing"],
    "Covariate": [
      "List values [1.0, 2.0, 3.9, 4.0, 5.0, 2.2]",
      "Threshold 0.3",
      "Pairwise comparison algorithm"
```

```
    ],
    "Mediator": ["Range iteration logic (0 <= i < j < len(numbers))"],
    "Outcome": ["True"]
  },
  "counterfactual_roles": {
    "Exposure": ["1-based indexing"],
    "Covariate": [
      "List values [1.0, 2.0, 3.9, 4.0, 5.0, 2.2]",
      "Threshold 0.3",
      "Pairwise comparison algorithm"
    ],
    "Mediator": ["Range iteration logic (1 <= i < j <= len(numbers))"],
    "Outcome": ["True"]
  },
  "causal_graph": {
    "factual_edges": [
      ["List values [...]", "0-based indexing"],
      ["Threshold 0.3", "0-based indexing"],
      ["Pairwise comparison algorithm", "0-based indexing"],
      ["0-based indexing", "Range iteration logic (0 <= i < j < len(
          numbers))"],
      ["Range iteration logic (0 <= i < j < len(numbers))", "True"],
      ["List values [...]", "True"],
      ["Threshold 0.3", "True"]
    ],
    "counterfactual_edges": [
      ["List values [...]", "1-based indexing"],
      ["Threshold 0.3", "1-based indexing"],
      ["Pairwise comparison algorithm", "1-based indexing"],
      ["1-based indexing", "Range iteration logic (1 <= i < j <= len(
          numbers))"],
      ["Range iteration logic (1 <= i < j <= len(numbers))", "True"],
      ["List values [...]", "True"],
      ["Threshold 0.3", "True"]
    ]
  }
}
```

**Open-Critic Example** This dataset performs a programming-related task: code generation. It is a synthetic code editing benchmark designed to train and evaluate large language models on their ability to identify and fix bugs in code. Each example consists of a natural language *task description*, a *correct code* solution that satisfies the task, and a *counterfactual explanation* that describes bugs introduced into a similar but faulty implementation. The objective is for the model to learn to generate or repair the *bugged version of the code* based on the bug descriptions as a counterfactual scenario. For instance, given a correct nested loop that prints all valid digit combinations excluding the number 5 and repetitions, the counterfactual explanation points out altered range values (e.g., using range(100) and range(1)), leading to an incomplete iteration logic. The full example is shown as follows:

```
{
    "task": "Create a nested loop to print every combination of numbers
        between 0-9, excluding any combination that contains the number
        5. Additionally, exclude any combination that contains a
        repeating digit. Implement the solution without using any built-
        in functions or libraries to check for repeating digits.",
    "correct_code": "for i in range(10):  # First digit\n    for j in
        range(10):  # Second digit\n  for k in range(10):  # Third digit\
        n    # Checking for the conditions\n   if i != 5 and j != 5 and k
         != 5 and i != j and i != k and j != k:\n   print(i, j, k)",
    "correct_explanation": "This code will generate and print every
        combination of three digits between 0-9 that do not contain the
        number 5 and do not have any repeating digits.",
```

```
    "bugged_code": "for i in range(10):  \n    for j in range(100):  \n
        for k in range(1):  \n   if i != 5 and j != 5 and k != 5 and i !=
         j and i != k and j != k:\n    print(i, j, k)",
    "counterfactual_explanation": "Reading through this code, I notice
        that:\n\n* The second loop is set to `range(100)` instead of `
        range(10)`, which means it will only iterate over the numbers
        from 0 to 99. This will cause the script to miss some
        combinations.\n* The third loop is set to `range(1)`, which means
         it will only iterate over the number 0. This will cause the
        script to only print combinations with one digit, instead of
        three.\n\nThese bugs will prevent the script from generating and
        printing all possible combinations of three digits between 0-9
        that do not contain the number 5 and do not have any repeating
        digits.\n\nTips for avoiding these mistakes:\n\n* Double-check
        the range values in each loop to ensure they are correct.\n* Make
         sure the loops iterate correctly over the desired range of
        values."
}
```

**Open-Critic Causality & Causal Graph.**

```
{
    "factual_roles": {
        "Exposure": ["Correct explanation (valid ranges and checks)"],
        "Covariate": [
          "Task requirements (0-9 digits)",
          "Exclusion logic (no 5/repeats)",
          "Nested loop structure"
        ],
        "Mediator": ["Proper range initialization (range(10) x3)"],
        "Outcome": ["Correct triple-nested loop code"]
      },
    "counterfactual_roles": {
        "Exposure": ["Counterfactual explanation (invalid ranges)"],
        "Covariate": [
          "Task requirements (0-9 digits)",
          "Exclusion logic (no 5/repeats)",
          "Nested loop structure"
        ],
        "Mediator": [
          "Flawed range parameters (range(100)/range(1))",
          "Incomplete digit iteration"
        ],
        "Outcome": ["Bugged code with limited iterations"]
        },
    "causal_graph": {
        "factual_edges": [
          ["Task requirements", "Correct explanation"],
          ["Exclusion logic", "Correct explanation"],
          ["Nested loop structure", "Correct explanation"],
          ["Task requirements", "Proper range initialization"],
          ["Exclusion logic", "Proper range initialization"],
          ["Nested loop structure", "Proper range initialization"],
          ["Correct explanation", "Correct triple-nested loop code"],
          ["Task requirements", "Correct triple-nested loop code"],
          ["Exclusion logic", "Correct triple-nested loop code"],
          ["Correct explanation", "Proper range initialization"],
          ["Proper range initialization", "Correct triple-nested loop
              code"],
          ["Correct explanation", "Correct triple-nested loop code"]
        ],
        "counterfactual_edges": [
          ["Task requirements", "Flawed range parameters"],
          ["Exclusion logic", "Flawed range parameters"],
          ["Nested loop structure", "Flawed range parameters"],
```

```
            ["Task requirements", "Bugged code with limited iterations"],
            ["Exclusion logic", "Bugged code with limited iterations"],
            ["Nested loop structure", "Bugged code with limited iterations"
                ],
            ["Counterfactual explanation", "Flawed range parameters"],
            ["Flawed range parameters", "Incomplete digit iteration"],
            ["Incomplete digit iteration", "Bugged code with limited
                iterations"],
            ["Counterfactual explanation", "Bugged code with limited
                iterations"]
        ]
    }
}
```

**Code-Preference Example** This dataset performs a programming-related task: code summarization. It contains pairs of duplicate code examples, with the only difference being the bugged code example has the bugged code 'surgically transplanted in' while the corrected code is left the same. Each example consists of a natural language *instruction*, a *correct code* solution that satisfies the instruction, and a *bug explanation* that describes bugs. The objective is for the model to learn to summarize and generate the bug descriptions as a counterfactual scenario. For instance, given a correct nested loop that prints all valid digit combinations excluding the number 5 and repetitions compared with a bugged loop, the bug description generated by the model will be able to point out altered range values (e.g., using range(100) and range(1)) leading to an incomplete iteration logic in its summarized response of bug explanation. The full example is shown as follows:

```
{
    "Instruction": 'Create a nested loop to print every combination of
        numbers between 0-9, excluding any combination that contains the
        number 5. Additionally, exclude any combination that contains a
        repeating digit. Implement the solution without using any built-
        in functions or libraries to check for repeating digits.',
    "bugged_code": 'What are the problems with this code? '''\npython\
        nfor i in range(10):  \n    for j in range(100):  \n for k in
        range(1):  \n if i != 5 and j != 5 and k != 5 and i != j and i !=
         k and j != k:\n  print(i, j, k)\n''',
    "bug_explanation": 'Reading through this code, I notice that:\n\n*
        The second loop is set to `range(100)` instead of `range(10)`,
        which means it will only iterate over the numbers from 0 to 99.
        This will cause the script to miss some combinations.\n* The
        third loop is set to `range(1)`, which means it will only iterate
         over the number 0. This will cause the script to only print
        combinations with one digit, instead of three.\n\nThese bugs will
         prevent the script from generating and printing all possible
        combinations of three digits between 0-9 that do not contain the
        number 5 and do not have any repeating digits.\n\nTips for
        avoiding these mistakes:\n\n* Double-check the range values in
        each loop to ensure they are correct.\n* Make sure the loops
        iterate correctly over the desired range of values.\n\nHere is
        the corrected code:\n\n```python\nfor i in range(10):  # First
        digit\n    for j in range(10):  # Second digit\n  for k in range
        (10):  # Third digit\n    # Checking for the conditions\n  if i
        != 5 and j != 5 and k != 5 and i != j and i != k and j != k:\n
            print(i, j, k)\n''',
    "correct_code": 'Here is an example of a nested loop in Python to
        print every combination of numbers between 0-9, excluding any
        combination that contains the number 5 or repeating digits:\n\n
        ```python\nfor i in range(10):  # First digit\n    for j in range
        (10):  # Second digit\n  for k in range(10):  # Third digit\n  #
        Checking for the conditions\n   if i != 5 and j != 5 and k != 5
        and i != j and i != k and j != k:\n    print(i, j, k)\n```\n\
        nThis code will generate and print every combination of three
        digits between 0-9 that do not contain the number 5 and do not
        have any repeating digits.'
}
```

**Code-Preferencec Causality & Causal Graph.**

```
{
  "factual_roles": {
    "Exposure": ["Correct code (range(10) loops)"],
    "Covariate": [
      "Task requirements (0-9 digits)",
      "Exclusion logic (no 5/repeats)",
      "Nested loop structure"
    ],
    "Mediator": ["Proper range initialization (range(10) x3)"],
    "Outcome": ["Correct explanation (valid ranges and checks)"]
  },
  "counterfactual_roles": {
    "Exposure": ["Bugged code (range(100)/range(1))"],
    "Covariate": [
      "Task requirements (0-9 digits)",
      "Exclusion logic (no 5/repeats)",
      "Nested loop structure"
    ],
    "Mediator": ["Flawed range parameters", "Incomplete digit iteration"]
      ,
    "Outcome": ["Bug explanation (incorrect ranges analysis)"]
  },
  "causal_graph": {
    "factual_edges": [
      ["Task requirements (0-9 digits)", "Correct code (range(10) loops)"
          ],
      ["Exclusion logic (no 5/repeats)", "Correct code (range(10) loops)"
          ],
      ["Nested loop structure", "Correct code (range(10) loops)"],
      ["Task requirements (0-9 digits)", "Proper range initialization (
          range(10) x3)"],
      ["Exclusion logic (no 5/repeats)", "Proper range initialization (
          range(10) x3)"],
      ["Nested loop structure", "Proper range initialization (range(10)
          x3)"],
      ["Task requirements (0-9 digits)", "Correct explanation (valid
          ranges and checks)"],
      ["Exclusion logic (no 5/repeats)", "Correct explanation (valid
          ranges and checks)"],
      ["Correct code (range(10) loops)", "Proper range initialization (
          range(10) x3)"],
      ["Proper range initialization (range(10) x3)", "Correct explanation
          (valid ranges and checks)"],
      ["Correct code (range(10) loops)", "Correct explanation (valid
          ranges and checks)"]
    ],
    "counterfactual_edges": [
      ["Task requirements (0-9 digits)", "Flawed range parameters"],
      ["Task requirements (0-9 digits)", "Incomplete digit iteration"],
      ["Task requirements (0-9 digits)", "Bug explanation (incorrect
          ranges analysis)"],
      ["Exclusion logic (no 5/repeats)", "Flawed range parameters"],
      ["Exclusion logic (no 5/repeats)", "Incomplete digit iteration"],
      ["Exclusion logic (no 5/repeats)", "Bug explanation (incorrect
          ranges analysis)"],
      ["Nested loop structure", "Flawed range parameters"],
      ["Nested loop structure", "Incomplete digit iteration"],
      ["Nested loop structure", "Bug explanation (incorrect ranges
          analysis)"],
      ["Bugged code (range(100)/range(1))", "Flawed range parameters"],
      ["Flawed range parameters", "Incomplete digit iteration"],
```

```
      ["Incomplete digit iteration", "Bug explanation (incorrect ranges
          analysis)"],
      ["Bugged code (range(100)/range(1))", "Bug explanation (incorrect
          ranges analysis)"]
    ]
  }
}
```

# B  COMPLEMENTARY EXPERIMENT

## B.1  SETTING

This section lists the experimental settings used in this study.

Table 7: LLM query hyperparameters used during all experiments.

| Hyperparameter | Value | Description |
|---|---|---|
| Temperature | 0.7 | Controls randomness in generation |
| Top-$p$ (nucleus sampling) | 0.95 | Probability mass for sampling |
| Max tokens | 2048 | Maximum number of tokens to generate |
| Stop sequences | ["\n", "Q:"] | Used to truncate responses |
| Prompt format | CoT, CoT-SC, ToT | Prompting strategy used in Section 4.2.2 |
| Tool-calling API | Enabled (Selective) | Used in tool-augmented experiments |

**Computational Resources.** All experiments were conducted on a high-performance computing server equipped with six NVIDIA RTX 6000 Ada Generation GPUs, each with 49 GB of dedicated VRAM. The system utilized CUDA version 12.8 and NVIDIA driver version 570.124.06. These GPUs supported parallel execution of model querying, evaluation, and tool-augmented tasks across our benchmark datasets. The hardware configuration ensured sufficient memory bandwidth and processing capability to accommodate large-scale inference, particularly for multimodal tasks and multi-sample prompting strategies such as CoT-SC and ToT. No resource-related constraints were encountered during experimentation.

## B.2  ADDITIONAL RESULT AND DISCUSSION

This part presents additional experimental results that complement the main evaluation in the body of the paper. These supplementary findings, together with what has been presented in previous sections, offer comprehensive insights into LLMs capabilities across different decompositional tasks.

To deepen our understanding of how LLMs handle counterfactual reasoning, we analyze representative datasets that cover text, multimodal, symbolic, and code-based modalities. We aim to uncover impediments at each decompositional stage. Below, we highlight dataset-level characteristics that either enable or hinder performance.

**Textual Question-Answering and Logic Parsing.** Datasets built on natural language (e.g., textual QA and logical modification tasks) generally facilitate the recognition of explicit causal variables such as exposures and outcomes. Models can easily link a stated intervention ("if tutoring was not provided") to its corresponding variable. However, mediators are often described abstractly (through latent constructs like "trust," "motivation," or "belief states") rather than explicit entities. This makes Task I disproportionately difficult for mediators compared to exposures or outcomes. Errors at this stage then propagate to Task IV, where the model must simulate how such mediators would affect outcomes under intervention. Even when causal graph construction (Task II) is near-perfect due to the structured nature of logical relations, the absence of explicit mediators limits downstream reasoning fidelity.

**Vision-Language Counterfactuals.** Multimodal datasets combining images with text pose unique challenges. When asked to identify causal variables, LLMs must ground textual descriptions in visual objects. For example, distinguishing "presence of a ball" (exposure) from "action of kicking" (mediator) requires fine-grained alignment of object attributes with causal semantics. This grounding

Table 8: LLMs' performance (F1 standard deviations, scaled to 100%) in causal variable identification (reordered; adjusted variability).

| **Dataset** | GPT-5 | | GPT-o4 | | Qwen3 | | Llama4-S | | Llama4-M | | Gemini2.5 | | DeepSeek | |
|---|---|---|---|---|---|---|---|---|---|---|---|---|---|---|
| | $v_1$ | $v_2$ | $v_1$ | $v_2$ | $v_1$ | $v_2$ | $v_1$ | $v_2$ | $v_1$ | $v_2$ | $v_1$ | $v_2$ | $v_1$ | $v_2$ |
| $v_1 = X$ (Exposure), $v_2 = Z$ (Covariate) | | | | | | | | | | | | | | |
| CRASS | 4.6 | 4.9 | 9.1 | 9.9 | 5.6 | 6.2 | 8.3 | 7.9 | 3.5 | 4.4 | 4.5 | 5.3 | 4.8 | 5.0 |
| CLOMO | 4.8 | 5.1 | 9.3 | 10.0 | 6.8 | 7.2 | 9.6 | 8.9 | 5.0 | 5.5 | 5.9 | 6.5 | 5.3 | 5.7 |
| RNN-Topo | 5.0 | 5.3 | 9.5 | 10.2 | 7.3 | 7.8 | 10.2 | 9.5 | 5.4 | 5.9 | 6.4 | 6.9 | 5.8 | 6.2 |
| CVQA-Bool | 5.4 | 5.7 | 9.8 | 10.4 | 9.7 | 10.2 | 12.4 | 13.1 | 8.9 | 9.4 | 10.8 | 11.5 | 9.4 | 10.1 |
| CVQA-Count | 5.6 | 5.9 | 9.9 | 10.5 | 10.6 | 11.3 | 13.7 | 14.2 | 9.8 | 10.5 | 11.5 | 12.3 | 10.3 | 11.2 |
| COCO | 5.8 | 6.1 | 10.0 | 10.6 | 11.9 | 12.6 | 15.2 | 15.8 | 11.1 | 11.8 | 12.9 | 13.7 | 11.6 | 12.5 |
| Arithmetic | 5.2 | 5.5 | 9.4 | 10.1 | 8.9 | 9.5 | 11.3 | 12.0 | 7.5 | 8.1 | 8.4 | 9.1 | 7.9 | 8.6 |
| MalAlgoQA | 5.3 | 5.6 | 9.6 | 10.3 | 9.5 | 10.1 | 12.2 | 12.9 | 8.2 | 8.7 | 9.0 | 9.7 | 8.5 | 9.2 |
| HumanEval-Exe | 6.0 | 6.3 | 10.2 | 10.8 | 12.8 | 13.6 | 16.3 | 17.1 | 11.7 | 12.4 | 13.9 | 14.7 | 12.5 | 13.4 |
| Open-Critic | 6.1 | 6.4 | 10.3 | 10.9 | 13.5 | 14.2 | 17.1 | 17.8 | 12.6 | 13.3 | 14.6 | 15.3 | 13.7 | 14.5 |
| Code-Preference | 6.0 | 6.2 | 10.1 | 10.7 | 13.1 | 13.9 | 16.7 | 17.4 | 12.2 | 12.9 | 14.3 | 15.0 | 13.1 | 13.9 |
| $v_1 = M$ (Mediator), $v_2 = Y$ (Outcome) | | | | | | | | | | | | | | |
| CRASS | 4.9 | 5.2 | 9.2 | 10.0 | 9.6 | 7.4 | 12.1 | 9.7 | 8.5 | 6.1 | 9.1 | 6.8 | 8.7 | 6.2 |
| CLOMO | 5.0 | 5.3 | 9.4 | 10.2 | 10.5 | 8.1 | 12.9 | 10.3 | 9.2 | 6.6 | 9.8 | 7.5 | 9.3 | 6.8 |
| RNN-Topo | 5.1 | 5.4 | 9.6 | 10.3 | 11.1 | 8.5 | 13.6 | 10.8 | 9.6 | 7.0 | 10.5 | 7.9 | 9.8 | 7.2 |
| CVQA-Bool | 5.7 | 6.0 | 10.0 | 10.6 | 13.8 | 11.3 | 16.9 | 14.2 | 12.4 | 10.0 | 14.1 | 12.5 | 13.2 | 10.7 |
| CVQA-Count | 5.9 | 6.2 | 10.1 | 10.7 | 14.6 | 12.1 | 17.8 | 15.1 | 13.3 | 10.8 | 15.0 | 13.3 | 14.1 | 11.6 |
| COCO | 6.0 | 6.3 | 10.2 | 10.8 | 15.3 | 12.8 | 18.5 | 15.9 | 14.0 | 11.5 | 15.7 | 14.1 | 14.8 | 12.4 |
| Arithmetic | 5.4 | 5.7 | 9.5 | 10.2 | 12.5 | 9.8 | 15.4 | 12.5 | 11.0 | 8.6 | 13.1 | 9.2 | 11.3 | 8.7 |
| MalAlgoQA | 5.6 | 5.9 | 9.7 | 10.4 | 13.1 | 10.4 | 16.1 | 13.2 | 11.6 | 9.2 | 13.8 | 10.0 | 12.0 | 9.3 |
| HumanEval-Exe | 6.2 | 6.5 | 10.4 | 11.0 | 16.2 | 13.6 | 19.3 | 16.7 | 14.8 | 12.3 | 16.5 | 14.8 | 15.5 | 13.1 |
| Open-Critic | 6.3 | 6.6 | 10.5 | 11.1 | 16.9 | 14.4 | 8.1 | 17.5 | 15.5 | 13.0 | 17.2 | 15.5 | 16.1 | 13.9 |
| Code-Preference | 6.1 | 6.4 | 10.3 | 10.9 | 16.5 | 14.0 | 19.7 | 17.0 | 15.2 | 12.6 | 16.9 | 15.1 | 15.8 | 13.5 |

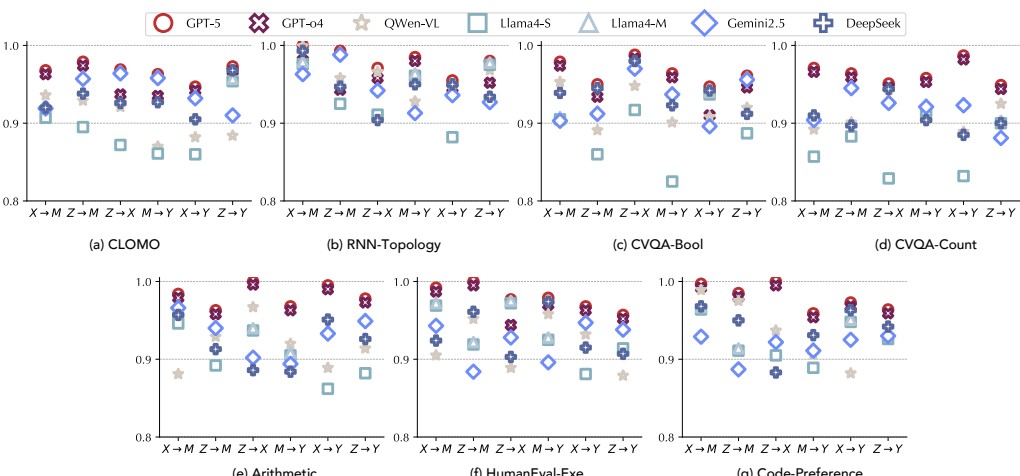

Figure 5: Additional evaluation on causal graph construction, complementing to Figure 3.

step introduces errors in Task I, especially when visual scenes are cluttered or ambiguous. Even when interventions (Task III) are identified correctly, outcome reasoning (Task IV) suffers because models struggle to propagate visual changes into numerical or behavioral predictions. For instance, recognizing that "removing a ball" should reduce the count of possible goals involves chaining visual detection, counting logic, and causal propagation—steps that current LLMs rarely integrate coherently.

**Symbolic and Mathematical Reasoning.** Datasets built on arithmetic or algorithmic transformations highlight another bottleneck: reliance on memorized patterns instead of causal mechanisms. In variable identification (Task I), explicit quantities are correctly recognized. In causal graph construction (Task II), rules linking operations (e.g., "base conversion influences the final number") are

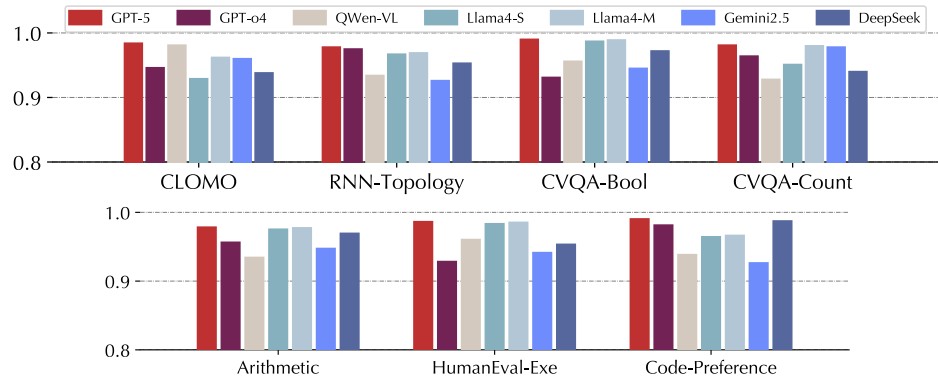

Figure 6: Additional evaluation of LLMs' intervention identification, complementing to Figure 4.

Table 9: LLM performance (F1 standard deviation) in reasoning the counterfactual mediator ($M'$) and outcome ($Y'$).

| Dataset | GPT-5 | | GPT-o4 | | Qwen3 | | Llama4-S | | Llama4-M | | Gemini2.5 | | DeepSeek | |
|---|---|---|---|---|---|---|---|---|---|---|---|---|---|---|
| | $M'$ | $Y'$ | $M'$ | $Y'$ | $M'$ | $Y'$ | $M'$ | $Y'$ | $M'$ | $Y'$ | $M'$ | $Y'$ | $M'$ | $Y'$ |
| CRASS | 4.2 | 5.0 | 4.8 | 5.6 | 6.7 | 8.5 | 10.0 | 12.5 | 5.0 | 7.1 | 6.3 | 8.1 | 5.5 | 7.4 |
| CLOMO | 4.4 | 5.2 | 5.0 | 5.9 | 7.2 | 9.0 | 10.6 | 13.0 | 5.4 | 7.5 | 6.7 | 8.6 | 6.0 | 7.9 |
| RNN-Topo | 4.6 | 5.4 | 5.1 | 6.0 | 7.8 | 9.5 | 11.0 | 13.5 | 5.6 | 7.7 | 7.2 | 9.1 | 6.5 | 8.4 |
| CVQA-Bool | 4.8 | 5.6 | 5.2 | 6.1 | 11.2 | 13.3 | 14.5 | 16.8 | 9.0 | 11.1 | 12.1 | 14.0 | 10.4 | 12.5 |
| CVQA-Count | 5.0 | 5.8 | 5.3 | 6.3 | 12.0 | 14.1 | 15.2 | 17.6 | 9.8 | 11.9 | 12.9 | 14.9 | 11.2 | 13.3 |
| COCO | 5.1 | 6.0 | 5.4 | 6.4 | 12.7 | 14.8 | 15.9 | 18.3 | 10.5 | 12.7 | 13.6 | 15.6 | 11.9 | 14.0 |
| Arithmetic | 4.5 | 5.2 | 5.1 | 6.0 | 9.5 | 11.6 | 12.8 | 15.3 | 7.4 | 9.5 | 9.0 | 11.1 | 8.3 | 10.4 |
| MalAlgoQA | 4.7 | 5.3 | 5.2 | 6.1 | 10.1 | 12.2 | 13.4 | 15.9 | 7.9 | 10.0 | 9.7 | 11.8 | 8.9 | 11.0 |
| HumanEval-Exe | 5.2 | 6.2 | 5.5 | 6.5 | 13.3 | 15.4 | 16.5 | 18.9 | 11.0 | 13.2 | 14.2 | 16.2 | 12.5 | 14.7 |
| Open-Critic | 5.3 | 6.3 | 5.6 | 6.6 | 13.9 | 16.0 | 17.0 | 19.6 | 11.6 | 13.8 | 14.8 | 16.9 | 13.1 | 15.3 |
| Code-Preference | 5.1 | 6.1 | 5.5 | 6.5 | 13.6 | 15.7 | 16.7 | 19.2 | 11.3 | 13.5 | 14.5 | 16.5 | 12.8 | 15.0 |

applied consistently. However, in outcome reasoning (Task IV), models frequently fail to simulate the correct causal pathway, often defaulting to template-based responses rather than computing the actual counterfactual result. This suggests that while symbolic data supports high precision in explicit structure, it exposes the weakness of LLMs in mechanistic simulation of causal processes.

**Code-Based Reasoning.** Programming-oriented datasets such as code execution, code generation, or preference tasks are particularly difficult across all stages. In Task I, identifying exposures and outcomes is hindered by the abstractness of programming constructs (e.g., "function signature" as exposure, "program output" as outcome). Mediators (such as intermediate execution states) are even harder to capture, as they are not explicitly represented in the code text but must be inferred from semantics. In Task II, while models can generate plausible causal graphs describing dependencies among variables or functions, these graphs often overgeneralize or miss critical execution details. Task IV is especially challenging: even when interventions like "changing a loop to recursion" are recognized, LLMs often fail to simulate downstream program behavior, producing outcomes that are logically plausible but incorrect. This reflects a persistent gap between syntactic recognition and semantic reasoning.

**Cross-Cutting Observations.** Across modalities, two key impediments recur:

(1) *Complex modalities impede variable identification.* Images and code introduce higher error rates in Task I, since grounding or semantic parsing must precede causal reasoning.

(2) *Implicit mediators bottleneck outcome reasoning.* Regardless of modality, when mediators are abstract or not explicitly present, performance in Task IV drops substantially. LLMs can identify interventions reliably, but they fail to propagate their effects along causal chains to yield consistent counterfactual outcomes.

These findings suggest that LLMs are "eligible" for decompositional reasoning in structured settings with explicit causal variables (e.g., clean text or arithmetic). However, when confronted with modality complexity or implicit causal pathways, their reasoning capacity is significantly impeded.

### B.3 Working Memory Perspective to Intepret Counterfactual Reasoning

Prior research (Zhang et al., 2024a) has demonstrated that language models exhibit notable difficulty in temporally storing and manipulating information even in n-back tasks that are cognitively simpler than explicit reasoning. This underlying limitation in working memory capacity poses constraints on long-term and multi-step reasoning. To explore the connection between memory bottlenecks and mediator identification challenges, we conducted additional experiments in more depth.

**Experiments on Working Memory.** To examine how working memory affects mediator reasoning, we designed a controlled n-back Mediator Recall task. While most benchmarks involve only single-step mediation, the Open-Critic dataset (code modality) includes examples of multi-step causal mediation. For instance, adjusting code inputs requires reasoning over prior inputs and transformations. In this task, the mediator must be inferred from causal variables presented n steps earlier in the input. We vary n from 1 to 3 and report F1 scores consistent with Table 10.

Table 10: LLM performance (F1) in n-back mediator recall

| Model | 1-hop | 2-hop | 3-hop |
|---|---|---|---|
| GPT-o4 | 72.2% | 63.5% | 9.7% |
| Qwen | 58.3% | 39.6% | 12.1% |
| Gemini | 66.4% | 26.1% | 7.5% |
| LLaMA4-Scout | 45.5% | 47.2% | 3.6% |

**Findings and insights.** These results reveal a sharp performance drop as the number of intermediate steps increases. From a working memory perspective, this suggests that current LLMs struggle to retain or reconstruct causal paths to mediators when they are separated by multiple reasoning hops. This degradation highlights a key constraint in long-horizon causal reasoning.

These findings align with our earlier observation that mediator reasoning is a consistent bottleneck in decompositional analysis. By framing this in terms of working memory capacity, we offer a mechanistic explanation for why LLMs falter on such tasks and why enhanced memory mechanisms (e.g., intermediate supervision or tool-assisted retrieval) may be necessary for progress.

### B.4 Influence of Model Scale

Table 11: LLM performance in reasoning variables $(X, Z, M, Y, M', Y')$.

| Dataset | Qwen-VL-2B | | | | | | Qwen-VL-4B | | | | | | Qwen-VL-8B | | | | | |
|---|---|---|---|---|---|---|---|---|---|---|---|---|---|---|---|---|---|---|
| | $X$ | $Z$ | $M$ | $Y$ | $M'$ | $Y'$ | $X$ | $Z$ | $M$ | $Y$ | $M'$ | $Y'$ | $X$ | $Z$ | $M$ | $Y$ | $M'$ | $Y'$ |
| CRASS | 44.2 | 27.1 | 20.3 | 37.2 | 18.2 | 16.5 | 47.0 | 29.5 | 22.1 | 39.4 | 20.3 | 18.7 | 52.8 | 34.4 | 26.7 | 44.5 | 25.8 | 23.6 |
| CLOMO | 23.8 | 18.4 | 11.6 | 15.3 | 8.3 | 5.1 | 26.0 | 20.2 | 13.0 | 17.0 | 9.6 | 6.4 | 30.5 | 24.0 | 16.5 | 20.8 | 12.5 | 9.3 |
| CVQA-Count | 23.1 | 16.2 | 15.6 | 18.4 | 11.5 | 7.4 | 25.4 | 18.0 | 17.1 | 20.3 | 13.1 | 8.9 | 29.7 | 21.3 | 20.4 | 23.8 | 16.4 | 11.8 |
| MalAlgoQA | 17.2 | 14.0 | 10.5 | 14.6 | 8.5 | 6.2 | 19.1 | 15.6 | 12.0 | 16.3 | 10.2 | 7.6 | 23.4 | 19.3 | 15.7 | 19.7 | 13.6 | 10.5 |

Besides comparing GPT and Llama families, we further conduct a controlled scaling study over the Qwen-VL series to examine how model size influences both causal variable identification and downstream counterfactual reasoning. Table 11 summarizes the performance of Qwen-VL-2B, 4B, and 8B across factual variables $(X, Z, M, Y)$ and counterfactual targets $(M', Y')$. Overall, we observe a clear but non-uniform scaling trend. Moving from 2B to 4B yields modest and consistent gains, particularly on explicit variables $(X, Z, Y)$, suggesting that moderate scaling primarily improves surface-level grounding and extraction. In contrast, the 8B model shows more noticeable improvements across both explicit and implicit variables, including the more challenging mediators $(M, M')$ and counterfactual outcomes $(Y')$. These gains indicate that larger Qwen models are better able to propagate interventions through the underlying causal structure rather than solely memorizing lexical

associations. However, even the 8B model retains substantial gaps on tasks requiring multi-hop causal reasoning, especially when mediators are implicit or visually grounded. Taken together, the scaling analysis suggests that increased model capacity helps alleviate some of the bottlenecks identified in smaller models, but does not by itself resolve the fundamental challenges of counterfactual mediation and outcome inference.

