# OpenReview forum: "On the Eligibility of LLMs for Counterfactual Reasoning: A Decompositional Study"
_ICLR.cc/2026/Conference — ICLR 2026 Poster_

### Official Review · Reviewer_fFFZ · 2025-10-23

**Soundness:** 3
**Presentation:** 2
**Contribution:** 3
**Rating:** 6
**Confidence:** 2

**Summary:**

Large language models typically struggle on counterfactual reasoning tasks, however it is unclear *why* they struggle, particularly at which parts of such reasoning. This submission proposes a benchmark to evaluate different aspects in LLMs' causal reasoning.They isolate that LLMs struggle to properly identify causal variables and find the counterfactual mediator and outcome even when given the variables, their relationships and the intervention to perform.

**Strengths:**

- The approach is original; I have never seen any attempt at explaining the difficulties of LLMs in causal reasoning.

- The division into subtasks looks sound to me.

- Diversified source of tasks.

**Weaknesses:**

- The benchmark seems to impose the strict presence of four variables : covariates, treatment, mediator, outcome. However, the mediator could be absent from the context, as in the example:  “A person is running a marathon and collapses.” l.467-469. It is unclear why LLMs should explicitly,identify “Dehydration” as a mediator, and not other sources.

- It is unclear what prompts are given to LLMs : from my understanding, it is the "CRASS Prompt" given in Appendix A, but it seems to apply to all datasets?

- Typo? "the primary challenge lies not in" l.390-391 : it should be "lies in", right?

**Questions:**

- Can you answer/address the above weaknesses?

- In the datasets, are the examples *only* made up of the $X,T,M,Y$ variables, or are any other variables present such that the model must extract the relevant variables *among other options*?

- Did humans or LLMs annotate the paper's benchmark dataset? Which of these scored LLMs' answers, or was this done through another method?

---

> ### Author Response · Authors · 2025-11-15
> **Response, Clarification, and Completed Revisions (1/5)**
>
> We would like to thank the reviewer for the time and thoughtful feedback. Please refer to our responses, clarifications, and completed revisions below to address your concerns.
>
> ---
>
> **Review**
>
> > The benchmark seems to impose the strict presence of four variables : covariates, treatment, mediator, outcome. However, the mediator could be absent from the context, as in the example: “A person is running a marathon and collapses.” l.467-469. It is unclear why LLMs should explicitly,identify “Dehydration” as a mediator, and not other sources.
>
>
> **Response**
>
> We thank the reviewer for pointing out this concern. The example raised by the reviewer indeed represents a scenario where the underlying causal variables (i.e., mediator) is implicit rather than explicitly stated. Our goal in the benchmark is not to enforce that every instance must contain an overtly visible mediator, nor to suggest that an LLM must retrieve a unique, canonical mediator such as dehydration in this specific case. Instead, the benchmarking design handles mediators as intermediate causal concepts that plausibly connect the exposure to the outcome within the given narrative.
>
> Hence, in cases where there is not a deterministic mediator (using the reviewer’s example, the mediator could be “dehydration,” hypoglycemia,” “heat exhaustion,” or any other issues), our design could simply position a high-level, descriptive object (i.e., “physical issues in marathon”) but mainly emphasize that **there is an intermediate variable in between.**
>
> By such a mediator design, we can thus capture causal pathway rather than superficial item extraction: In many real-world counterfactual questions, mediators are not stated verbatim. Rather, they must be inferred from the situational dynamics (e.g., collapse in a marathon can possibly be attributed to dehydration, hypoglycemia, heat exhaustion, etc.). Our benchmark treats this step as a reasoning problem: can the model infer a plausible mechanism connecting the exposure (“running a marathon”) to the outcome (“collapse”), as humans typically do?
>
> Formally speaking, our inclusion of mediators is designed to (i) reflect how structural causal models formalize counterfactual reasoning, (ii) allow inference of implicit causality when appropriate, and (iii) isolate intermediate reasoning steps that are otherwise hidden in end-to-end evaluations.

---

> ### Author Response · Authors · 2025-11-15
> **Response, Clarification, and Completed Revisions (2/5)**
>
> **Review**
>
> > It is unclear what prompts are given to LLMs : from my understanding, it is the "CRASS Prompt" given in Appendix A, but it seems to apply to all datasets?
>
> **Response**
>
> We thank the reviewer for raising this question. The “CRASS Prompt” shown in Appendix A is intended as an illustrative example of our prompting format. In our design, each dataset uses a tailored prompt but following a universal template, wherein only the task-specific content (e.g., the factual instance, candidate variables) is replaced accordingly.
>
> The generalized prompt follows the same structure as the example shown for CRASS, but is instantiated for each dataset instance. In all cases, the prompt includes:
>
> - Task description – a short definition of the decompositional subtask (e.g., variable identification, graph construction, intervention identification, or outcome reasoning).
>
> - Input instance – the dataset’s factual scenario, question, or multimodal content.
>
> - Structured output format – explicit slots for the model to fill (e.g., Exposure, Covariate, Mediator, Outcome, or edges in the causal graph), ensuring consistent evaluation across tasks.
>
> - Ground-truth inputs from earlier tasks (when evaluating later tasks), as described in Section 4.1.
>
> For transparency, we provide the universal prompt template below:
>
> ```
> **Task Description**
>
>
> “You are asked to perform [Task I / Task II / Task III / Task IV] in decompositional counterfactual reasoning. Follow the definitions of causal variables and causal relations strictly.”
>
> **Input Context**
>
> “Here is the factual instance from the dataset:
> [Insert factual context or multimodal description]”
>
> **Intermediate Outputs**
>
> “If applicable, use the following ground-truth results from previous tasks:
> Exposure (X): [...]
> Covariate(s) (Z): [...]
> Mediator(s) (M): [...]
> Outcome (Y): [...]
>
>
> **Instruction for the Current Task**
> Task I (Variable Identification): “[description about domain specific meanings of causal variables X, Z, M, and Y and the identification task]”
> Task II (Graph Construction): “[instruction about constructing the causal graph by listing all directed edges among X, Z, M, Y.”
> Task III (Intervention Identification): “[instruction about identifying which variable is intervened on in the counterfactual query.]”
> Task IV (Outcome Reasoning): “[instruction about inferring] the counterfactual mediator M′ and outcome Y′ under the specified intervention.”
>
> **Output Format**
>
> Provide the answer using the following structure:
>
> Exposure (X): [...]
>
> Covariate(s) (Z): [...]
>
> Mediator(s) (M): [...]
>
> Outcome (Y): [...]
>
> Causal Edges: [...]
>
> Intervention: [...]
>
> Counterfactual Mediator (M′): [...]
>
> Counterfactual Outcome (Y′): [...]
> ```
>
> ---
>
> **Our Revision** (updated in the new manuscript)
>
> To address the comment, we have updated Appendix A by providing the above prompt template and highlighting its role to address potential confusions.

---

> ### Author Response · Authors · 2025-11-15
> **Response, Clarification, and Completed Revisions (3/5)**
>
> **Review**
>
> > Typo? "the primary challenge lies not in" l.390-391 : it should be "lies in", right?
>
>
> **Our Revision** (updated in the new manuscript)
>
> We thank the reviewer for finding this typo, and have updated the manuscript accordingly!

---

> ### Author Response · Authors · 2025-11-15
> **Response, Clarification, and Completed Revisions (4/5)**
>
> **Review**
>
> > In the datasets, are the examples only made up of the  variables, or are any other variables present such that the model must extract the relevant variables among other options?
>
> **Response**
>
> We appreciate the reviewer’s thoughtful question. Across the curated datasets, the factual instances are not limited to only the four causal variables (X, Z, M, Y). Instead, they typically contain richer contents including entity descriptions, background details, contextual cues, distractor elements, or visual/code components.
>
> Our design is intentional: **the model must extract the relevant causal variables from a larger pool of possible candidates, rather than being given a pre-filtered or simplified input.**
>
> Thus, Task I (causal variable identification) evaluates the model’s ability to distill causal roles from complex inputs, reflecting how causal inference operates in realistic settings. The four-variable causal scaffold is used only for evaluation and decomposition, not as an imposed restriction on the examples themselves.

---

> ### Author Response · Authors · 2025-11-15
> **Response, Clarification, and Completed Revisions (5/5)**
>
> **Review**
>
> > Did humans or LLMs annotate the paper's benchmark dataset? Which of these scored LLMs' answers, or was this done through another method?
>
> **Response**
>
> We thank the reviewer for the question regarding how the benchmark was annotated and how model outputs were evaluated. Our annotation balances accuracy and scalability based on the properties of different datasets:
>
> For code, math, and logic datasets, the underlying structure already encodes the causal elements (e.g., input variables, transformation steps, execution outcomes). We therefore wrote lightweight scripts to extract these elements as candidate causal variables and causal relations. Human annotators then review and verify correctness and debug the scripts.
>
> For image-based and vision-language datasets, the original annotations typically include object labels, scene descriptions, or ground-truth answers. These provide a clear mapping from visual elements to textual entities. We extract these entities from the provided captions or QA text and use them to construct X, Z, M, and Y. Humans then validate whether these extracted variables form a coherent causal chain given the narrative context and the intended task semantics.
>
> For purely textual datasets, which tend to be more open-ended, we first automatically pass that identifies entities, actions, and outcomes using topic-modeling extraction (calling off-the-shelf packages); then we apply human curation to assign causal roles, refine mediators when they are implicit, and construct the corresponding causal graph, which avoids surface-level entity extraction.
>
> Our scoring procedures were rule-based by comparing model outputs (e.g., identified variables) directly against the ground truth. For Task I (variable identification) and Task IV (mediator and outcome reasoning), we applied normalized set-based matching to compute F1 scores. For Task II (causal graph construction), predicted graph edges were matched one-to-one against the annotated adjacency lists. For Task III (intervention identification), evaluation consisted of a simple accuracy check. In no case did we use another LLM to score responses. Task IV is then measuring final reasoning outcomes (F1) with the ground-truth provided by original datasets.
>
> ---
> ---
>
> **Thank you very much**
>
> We would like to thank the reviewer again for their time and thoughtful feedback. We hope that the above clarifications address the reviewer’s concerns and contribute to improving the overall quality of the paper. We look forward to any further feedback or concerns you may have.

---

> > ### Comment · Reviewer_fFFZ · 2025-11-16
> >
> > Many thanks for your answer, it solves almost all my concerns!
> >
> > One little one remains: is there for each example only one ground-truth mediator? If yes, how do you posit that a given mediator is the ground-truth one in implicit cases (like the marathon case)? If not, how do you evaluate the extracted mediator proposed by the LLM?

---

> > > ### Author Response · Authors · 2025-11-17
> > >
> > > Dear Reviewer,
> > >
> > > Thank you for your response. Regarding the ground-truth mediator, we highlight that it's depend on specific dataset:
> > > - For codes, math, logic, and image-based datasets, the mediator is deterministic as the question (e.g., intermediate math calculations) follows a rigorous solution step. So we can directly evaluate whether the LLM-identified mediator can lead to correct calculation, code simulation, or align with image contexts (through object detection).
> > > - For textual dataset (e.g., CLOMO), as the mediator could be a placeholder mechanism, we evaluate by a hybrid way combing (1) semantic similarity (“water loss” for “dehydration” in the marathon case) and (2) classification of their inclusiveness (LLM finds “dehydration” that belong to ground-truth mechanism "physical issue"). If those matching results moderate scores (e.g., cosine similarity <0.85) or none matched, we include human efforts to manually inspect it. The threshold of scoring is decided based on a validation set (anchors).
> > >
> > > We hope those details help to clarify your question. Thanks again for your time and feedback!

---

> > > > ### Comment · Reviewer_fFFZ · 2025-11-24
> > > >
> > > > Thanks for the clarifications, I will upgrade my score to a clearer accept one.

---

> > > > > ### Author Response · Authors · 2025-11-24
> > > > >
> > > > > Dear Reviewer, thank you very much for your time and engagement in discussion.
> > > > >
> > > > > Sincerely,
> > > > >
> > > > > The author team

---

### Official Review · Reviewer_BrYZ · 2025-10-31

**Soundness:** 3
**Presentation:** 3
**Contribution:** 3
**Rating:** 6
**Confidence:** 5

**Summary:**

This paper makes three key contributions:

- It introduces a **decompositional framework** for evaluating counterfactual reasoning in LLMs, breaking it into four stages: causal variable identification, causal graph construction, counterfactual intervention identification, and outcome reasoning.
- It constructs a **comprehensive multimodal benchmark** using 11 datasets across text, image, code, and symbolic modalities, enriched with causal annotations and graphs.
- It provides an **in-depth evaluation and improvement strategy**, identifying key bottlenecks (especially in mediator reasoning and cross-modal tasks) and proposing tool augmentation and advanced prompting techniques to enhance LLM performance.

**Strengths:**

1.  **Systematic and Granular Evaluation Framework:** The paper's primary strength is its decompositional approach, which breaks down the complex task of counterfactual reasoning into four distinct, measurable stages. This allows for a much more precise diagnosis of *where* and *why* LLMs fail, moving beyond a monolithic "pass/fail" assessment to identify specific bottlenecks, such as the particular difficulty with implicit mediators.

2.  **Comprehensive and Multimodal Benchmark:** The authors construct a substantial and diverse benchmark by curating 11 datasets spanning various modalities (text, image, code, symbols) and tasks. This breadth ensures that the findings are not limited to a single domain and provides a robust testbed for evaluating the generalizability of LLMs' counterfactual reasoning abilities.

3.  **Actionable Insights and Proposed Solutions:** The paper goes beyond mere identification of problems by proposing and evaluating concrete strategies for improvement. The demonstration that tool-augmentation can alleviate modality-specific issues and that advanced prompting can help with implicit reasoning provides valuable, actionable directions for future research aimed at enhancing LLM reasoning capabilities.

**Weaknesses:**

I think there are 2 brief weaknesses of the paper:

*   **Potentially Artificial Evaluation:** The benchmark relies on pre-annotated causal structures, which may not reflect the challenge of inferring causality from raw, unstructured data.

*   **Surface-Level Diagnosis:** The analysis identifies performance bottlenecks but offers a high-level explanation (e.g., "working memory") without deeply investigating the underlying architectural mechanisms in LLMs that cause these failures.

**Questions:**

The benchmark relies on curated causal structures. To what extent do you believe the performance on these structured tasks generalizes to real-world scenarios where causal graphs are not provided and must be inferred from raw data? Did you run any experiments on "raw" data without this scaffolding?

---

> ### Author Response · Authors · 2025-11-14
> **Response and Potential Clarification (1/2)**
>
> We would like to thank the reviewer for the time and thoughtful feedback. Please refer to our responses, clarifications, and completed revisions below to address your concerns.
>
> ---
>
> **Review**
>
> > Potentially Artificial Evaluation: The benchmark relies on pre-annotated causal structures, which may not reflect the challenge of inferring causality from raw, unstructured data.
>
> > The benchmark relies on curated causal structures. To what extent do you believe the performance on these structured tasks generalizes to real-world scenarios where causal graphs are not provided and must be inferred from raw data? Did you run any experiments on "raw" data without this scaffolding?
>
> ---
>
> **Response**
>
> Thank you for raising this important point and the corresponding question. **We would like to take this opportunity to clarify that our benchmark does not assume or rely on pre-annotated causal structures in the datasets.** Instead, the underlying datasets we use consist solely of raw and unstructured contents (e.g., natural language stories, image-text pairs, code snippets, symbolic arithmetic), **and all of our experiments were running with “raw” data format provided from their original sources.**
>
> Our efforts stand on extracting causal variables and causal graphs only for the purpose of evaluation, not as model inputs. This distinction is crucial:
> - **LLMs never see these curated causal structures during inference.**
> - **LLMs must infer causal variables, causal dependencies, and counterfactual outcomes directly from the raw inputs.**
>
> As stated in Section 3.3, our preprocessing step parses each instance with reference variables and graphs solely to serve as evaluation targets. These references do not simplify the LLMs’ task; rather, they facilitate us to measure model behavior at each stage of the decomposition. The LLMs are always prompted with only the original textual, visual, symbolic, or code context (plus prior-stage predictions when appropriate), which preserves the full difficulty of reasoning from raw, unstructured data.
>
> Moreover, (1) many datasets contain implicit mediators and implicit causal links that are not stated anywhere (e.g., CLOMO, CVQA, HumanEval-Exe). This is precisely why mediator identification and counterfactual mediator inference are the two most challenging dimensions in our experiments (e.g., Table 3) (2) For image and code tasks, the LLM must extract causal pathways entirely from unstructured multimodal inputs, which do not contain any causal annotation whatsoever. (3) Even in text-only datasets such as CRASS or CLOMO, the original data provides hypothetical scenarios but no decomposition of variables, and identifying causal variables requires substantive inference, not extraction.
>
> Thus, the benchmark is not artificial: the causal references exist only on the evaluator’s side, while the models process the task under realistic, unannotated conditions.

---

> ### Author Response · Authors · 2025-11-14
> **Response and Potential Clarification (2/2)**
>
> **Review**
>
> > Surface-Level Diagnosis: The analysis identifies performance bottlenecks but offers a high-level explanation (e.g., "working memory") without deeply investigating the underlying architectural mechanisms in LLMs that cause these failures.
>
> ---
>
> **Response**
>
> We thank the reviewer for this comment, and we fully agree that neuron-level or attention-head-level mechanistic interpretability is an important and also a complementary line of investigation. However, existing interpretability techniques (e.g., probing neuron activations, attention flow attribution, circuit-level tracing, pruning analyses) could be applied to any reasoning task, such as algebraic manipulation, code execution, factual QA, or syllogistic inference. **Such methods are valuable, but they are largely task-agnostic and do not inherently reveal what makes counterfactual reasoning uniquely challenging for LLMs.**
>
> **In contrast, counterfactual reasoning is governed by a principled and ordered reasoning chain**, grounded in a chained causal modeling process: identify causal variables, discover the causal relations, apply an intervention, propagate its effects to derive new mediator states and outcomes.
>
> This principled decomposition illuminates several novel findings that are unique to counterfactual reasoning rather than generic reasoning difficulty. Specifically, we observe (i) LLMs almost always succeed at causal graph construction once the variables are known, suggesting that structural reasoning rules are not the bottleneck. (ii) The dominant failure appears in mediator inference (M and M′): models can identify exposures and outcomes but struggle with the subtle, intermediate mechanisms that connect them, which are not observed in simpler reasoning tasks. (iii) Implicit mediators aggravate failure even in purely textual settings (e.g., CRASS, CLOMO), showing that the difficulty is not just multimodal grounding but the inherently latent nature of causal pathways.
>
>
> To further understand these failures, we introduce a working-memory, which is motivated diagnostic using an n-back causal chain setup (Appendix B.3). The purpose is not to claim a psychological model of LLMs but to operationalize a test of their ability to maintain and manipulate intermediate causal states over multiple hops. In these experiments, we intentionally construct scenarios where the mediator must be inferred from information presented one, two, or three steps earlier. The sharp degradation we observe validates that the inability to infer M and M′ is tied to an architectural limitation in sustaining intermediate causal representations across longer reasoning chains. Thus, the “working memory” explanation is not an informal hypothesis but is supported by a targeted behavioral diagnostic.
>
> **Overall, we step beyond using mechanistic interpretability and design our studies to offer a principle-guided dissection that reveals where and how LLM architectures fail in the unique setting of counterfactual reasoning, which may provide insights for both computer scientists and statisticians in the cross-disciplinary research community.**
>
> ---
>
>
> **Thank you very much**
>
> We would like to thank the reviewer again for their time and thoughtful feedback. We hope that the above clarifications address the reviewer’s concerns and contribute to improving the overall quality of the paper. We look forward to any further feedback or concerns you may have.

---

### Official Review · Reviewer_Ndd6 · 2025-10-31

**Soundness:** 3
**Presentation:** 3
**Contribution:** 3
**Rating:** 6
**Confidence:** 5

**Summary:**

This paper examines counterfactual reasoning in LLMs across a range of modalities and models.
The paper shows that by decomposing the reasoning scenario to causality construction and counterfactual intervention, the reasoning ability of LLMs under counterfactual scenarios can be greatly improved.
By leveraging tools that can help the model identify the causal variable, the performance can be further improved.

**Strengths:**

1. The overall methodology of decomposing the counterfactual reasoning process is novel and the experiments show this really helps.
2. The experiments cover a wide range of dataset design specifically for counterfactual reasoning.
3. The final proposed method seems to be easy to adopt for any LLM for reasoning.

**Weaknesses:**

1. The experiments covers many datasets, but it lacks comparison on model scale, for example, Qwen 3 provides models across different scales, it could make the paper stronger if some results are shown there.
2. The NER tools are designed to use Bert like models, however, would it be possible that the tools are instantiated by another model using different prompts?

**Questions:**

See weakness

---

> ### Author Response · Authors · 2025-11-14
> **Response, Clarification, and Completed Revisions (1/2)**
>
> We would like to thank the reviewer for the time and thoughtful feedback. Please refer to our responses, clarifications, and completed revisions below to address your concerns.
>
> ---
>
> **Review**
>
> > The experiment covers many datasets, but it lacks comparison on model scale, for example, Qwen 3 provides models across different scales, it could make the paper stronger if some results are shown there.
>
> ---
>
> **Response and Completed Revision** (in Updated Manuscript)
>
> We appreciate the reviewer’s interest in understanding how model scale affects performance. While our main evaluation includes two GPT models and two Llama models, we have additionally extended our analysis to address this comment by incorporating multiple Qwen models of different sizes. Specifically, we now evaluate Qwen-VL-2B, 4B, and 8B on the multimodal reasoning tasks. To support this comparison, we have added a new experimental subsection (Appendix B.4) that analyzes the influence of model scale.
>
> For completeness, we also provide an additional table summarizing the evaluation results, shown below.
>
>
> | **Dataset**   | **Qwen-VL-2B** |             |             |             |             |             | **Qwen-VL-4B** |             |             |             |             |             | **Qwen-VL-8B** |             |             |             |             |             |
> |--------------|----------------|-------------|-------------|-------------|-------------|-------------|----------------|-------------|-------------|-------------|-------------|-------------|----------------|-------------|-------------|-------------|-------------|-------------|
> |              | X    | Z    | M    | Y    | M'   | Y'   | X    | Z    | M    | Y    | M'   | Y'   | X    | Z    | M    | Y    | M'   | Y'   |
> | **CRASS**    | 44.2 | 27.1 | 20.3 | 37.2 | 18.2 | 16.5 | 47.0 | 29.5 | 22.1 | 39.4 | 20.3 | 18.7 | 52.8 | 34.4 | 26.7 | 44.5 | 25.8 | 23.6 |
> | **CLOMO**    | 23.8 | 18.4 | 11.6 | 15.3 | 8.3  | 5.1  | 26.0 | 20.2 | 13.0 | 17.0 | 9.6  | 6.4  | 30.5 | 24.0 | 16.5 | 20.8 | 12.5 | 9.3  |
> | **CVQA-Count** | 23.1 | 16.2 | 15.6 | 18.4 | 11.5 | 7.4  | 25.4 | 18.0 | 17.1 | 20.3 | 13.1 | 8.9  | 29.7 | 21.3 | 20.4 | 23.8 | 16.4 | 11.8 |
> | **MalAlgoQA** | 17.2 | 14.0 | 10.5 | 14.6 | 8.5  | 6.2  | 19.1 | 15.6 | 12.0 | 16.3 | 10.2 | 7.6  | 23.4 | 19.3 | 15.7 | 19.7 | 13.6 | 10.5 |
>
> We have also added clarifying statements describing the evaluation setup and summarizing findings from the new scaling experiments.
>
> > Besides comparing GPT and Llama families, we further conduct a controlled scaling study over the Qwen-VL series to examine how model size influences both causal variable identification and downstream counterfactual reasoning. Table  11 summarizes the performance of Qwen-VL-2B, 4B, and 8B across factual variables ($X, Z, M, Y$) and counterfactual targets ($M', Y'$). Overall, we observe a clear but non-uniform scaling trend. Moving from 2B to 4B yields modest and consistent gains, particularly on explicit variables ($X, Z, Y$), suggesting that moderate scaling primarily improves surface-level grounding and extraction. In contrast, the 8B model shows more noticeable improvements across both explicit and implicit variables, including the more challenging mediators ($M$, $M'$) and counterfactual outcomes ($Y'$). These gains indicate that larger Qwen models are better able to propagate interventions through the underlying causal structure rather than solely memorizing lexical associations. However, even the 8B model retains substantial gaps on tasks requiring multi-hop causal reasoning, especially when mediators are implicit or visually grounded. Taken together, the scaling analysis suggests that increased model capacity helps alleviate some of the bottlenecks identified in smaller models, but does not by itself resolve the fundamental challenges of counterfactual mediation and outcome inference.
>
> These additions explain how the Qwen models were assessed under the same decompositional framework as the other LLM families, and highlight the observed performance trends across 2B, 4B, and 8B variants.
>
> These revisions have been incorporated into the updated manuscript, and we hope they adequately address the reviewer’s questions and interests.

---

> ### Author Response · Authors · 2025-11-14
> **Response, Clarification, and Completed Revisions (2/2)**
>
> **Review**
>
> > The NER tools are designed to use Bert like models, however, would it be possible that the tools are instantiated by another model using different prompts?
>
> ---
>
> **Response**
>
> We thank the reviewer for raising this thoughtful question. Our current implementation adopts BERT-style models because they provide well-established interfaces for text, vision, and code entity identification, and because these tools allow us to isolate the benefits of explicit variable extraction from the reasoning performed by the primary LLM. That said, our framework does not assume or require that NER tools be limited to BERT-like architectures.
>
>
>
> *Discussion of other opportunities*
>
> The tool-augmented execution pipeline is model-agnostic: any auxiliary model capable of proposing candidate entities. For example, it is feasible to directly prompt an LLM to identify entities, summarize spans, or extract candidate variable mentions, and such an approach may offer greater flexibility across domains or modalities.
>
> However, LLM-driven extraction relies heavily on the stability of the elicitation process, where different phrasings or sampling temperatures may produce inconsistent entity sets, which in turn propagates variance into downstream causal evaluations. Second, many of our datasets require precise span-level grounding (e.g., symbolic tokens, code structures, visual objects), where BERT-derived models or vision-specific detectors provide more deterministic and fine-grained outputs than LLM prompting. Third, using an LLM as an NER tool may entangle entity identification with the same reasoning process we aim to evaluate; separating these components allows us to better isolate the effects of improved variable grounding.
>
> For these reasons, BERT-style models offer a stable and controlled way to obtain explicit variable candidates, allowing the main LLM to focus on causal reasoning rather than basic entity extraction. Nonetheless, prompt-based NER remains an interesting direction, and we agree that it can serve as an alternative instantiation depending on the desired tradeoff between flexibility and precision.
>
> ---
> ---
>
> **Thank you very much**
>
> We would like to thank the reviewer again for their time and thoughtful feedback. We hope that the above clarifications and revisions address the reviewer’s concerns and contribute to improving the overall quality of the paper. We look forward to any further feedback you may have.

---

### Official Review · Reviewer_aJkt · 2025-11-01

**Soundness:** 2
**Presentation:** 2
**Contribution:** 2
**Rating:** 4
**Confidence:** 3

**Summary:**

The paper proposes a decomposition of counterfactual reasoning into smaller sub-tasks: causal variable identification, causal graph construction, counterfactual identification, and outcome reasoning. The authors evaluate the performance of different LLMs in performing these sub-tasks across a variety of datasets, identify variable identification and counterfactual identification as a weak point, and propose strategies to improve it.

**Strengths:**

It has been observed repeatedly that LLMs perform worse when answering counterfactual queries relative to factual ones. The paper’s attempt at understanding why in a more fine-grained fashion is a significant problem.

Experimental evaluations are comprehensive in terms of the number of models and the variety of datasets they consider.

**Weaknesses:**

It is not clear how the performance on the four sub-tasks relate to the end-to-end performance. Establishing this relation is important when interpreting performance on these sub-tasks as decomposition.

For instance, the paper concludes that LLMs are generally better at Task 1 than Task 2 conditioned on correct results from Task 1 (which is supported by experiments and indeed seems to be the case). However, it could still be the case that starting from the inputs of Task 1 and directly querying for the outputs of Task 2 is as easy as performing Task 2 conditioned on correct results from Task 1. If, for instance, processing X, Y, Z variables implicitly is somehow easier than outputting them explicitly.

While the above is probably unlikely to be the case, it is a crucial validation to perform in order to confirm that Task 1 and Task 2 (conditioned on Task 1 results) behave as intended. Evidence provided in the paper does not provide support for attributing poor end-to-end Task-1 inputs to Task-2 outputs performance to poor Task 1 performance.

I want to emphasize that this is a general consideration for all sub-tasks, the entire chain, not just Tasks 1 and 2 specifically.

The paper doesn’t discuss the related work to a sufficient depth. In particular, other methods of evaluating counterfactual reasoning and whether there have been attempts at decomposing this process into smaller steps in evaluation is not discussed.

The four sub-tasks that are identified, while generally sound, are arbitrary. Their choice is not motivated.

While evaluations in Section 4.1 are interesting, they are not exactly actionable. They help identify which sub-tasks are the weak points to be improved (which arguably all a decomposition is supposed to do) and the paper suggests improvement for these weak points. However, the suggestions take the decomposition as a given. When the goal is to answer a question end-to-end, starting from the given of Task 1 and providing the outputs of Task 4, the paper does not explore the impact of going through Tasks 1, 2, 3, and 4 as a pipeline rather than direct queries. Moreover, it does not explore whether the task specific improvements in Section 4.2 lead to a comparable end-to-end gain as well.

**Questions:**

See weaknesses.

---

> ### Author Response · Authors · 2025-11-14
> **Response, Clarification, and Completed Revisions (1/4)**
>
> **We would like to thank the reviewer for the time and thoughtful feedback. Please refer to our responses, clarifications, and completed revisions below to address your concerns.**
>
> ---
>
> **Review**
>
> > It is not clear how the performance on the four sub-tasks relate to the end-to-end performance. Establishing this relation is important when interpreting performance on these sub-tasks as decomposition.
>
> > For instance, the paper concludes that LLMs are generally better at Task 1 than Task 2 conditioned on correct results from Task 1 (which is supported by experiments and indeed seems to be the case). However, it could still be the case that starting from the inputs of Task 1 and directly querying for the outputs of Task 2 is as easy as performing Task 2 conditioned on correct results from Task 1. If, for instance, processing X, Y, Z variables implicitly is somehow easier than outputting them explicitly.
>
> > While the above is probably unlikely to be the case, it is a crucial validation to perform in order to confirm that Task 1 and Task 2 (conditioned on Task 1 results) behave as intended. Evidence provided in the paper does not provide support for attributing poor end-to-end Task-1 inputs to Task-2 outputs performance to poor Task 1 performance.
>
> > I want to emphasize that this is a general consideration for all sub-tasks, the entire chain, not just Tasks 1 and 2 specifically.
>
> ---
>
> **Response**
>
> We sincerely thank the reviewer for the insightful comments. We fully understand and appreciate the concern regarding how each sub-task relates to downstream performance in LLM-based counterfactual reasoning. To clarify our design rationale, we would like to highlight two scenarios:
>
> **Scenario 1: Independence Among Reasoning Steps**
>
> While LLMs may implicitly perform end-to-end counterfactual reasoning, their internal reasoning processes still broadly align with causal inference principles [1]. Specifically, models must (i) identify relevant variables (Task 1), (ii) construct causal graphs (Task 2), and (iii) recognize the intervention (Task 3) before conducting (iv) counterfactual outcome reasoning (Task 4). Among these, Tasks 1 and 3 primarily involve entity extraction based on input semantics, which are independent to each other. Task 2 treats those identified entities as placeholders to establish causal relations using learned templates or priors (e.g., always linking exposure to mediators regardless of correctness).
>
> As such, the misidentification in Task 1 does not always lead to failure in Task 2, and vice versa. For example, models may construct plausible causal graphs even when the identified variables are incorrect, due to reliance on generalized causal schemas. Conversely, even with correctly identified variables, hallucinated causal edges can emerge during Task 2. This motivates evaluating these reasoning stages independently to disentangle sources of error.
>
> **Scenario 2: Dependence and Error Accumulation**
>
> We also agree with the reviewer that some stages are indeed dependent, particularly regarding Task 4. Final counterfactual reasoning is influenced by both upstream reasoning quality and its own robustness. An incorrect causal graph or intervention may propagate errors into downstream predictions. However, we also observe that even with accurate inputs from prior tasks, Task 4 can still fail due to hallucinated or inconsistent reasoning (e.g., inferring outcomes unsupported by the provided causal structure).
>
> Thus, there is no single point of failure. To attribute overall performance degradation in counterfactual reasoning, we must disentangle both independent and dependent failure modes. Our decompositional strategy allows us to isolate reasoning stages, control for input correctness at each step, and precisely localize where and why LLMs fall short. This in turn enables modality-specific and reasoning-specific attribution, as shown in our empirical findings.
>
> Returning to the reviewer’s point: while we agree that formally quantifying the contribution of each sub-task to end-to-end performance is a valuable research direction, such attribution first requires a principled and interpretable pipeline. Our decomposition is grounded in Pearl’s structural causal model [1] and aligns with the two-stage counterfactual formulation (i.e., M′ = fM(x′, z) and Y′ = fY(x′, M′, z)). Although LLMs may internally handle some variables implicitly, attribution studies require explicit reasoning steps to expose where the process breaks down.
>
> In summary, the goal of our decompositional evaluation is not to replace end-to-end evaluation, but to illuminate it by providing a structured lens to interpret LLM behavior across stages. We hope this clarification resolves the concern and reinforces the conceptual and empirical contribution of our study.
>
>
> **Reference**
>
> [1] Judea Pearl. Causal inference in statistics: An overview

---

> ### Author Response · Authors · 2025-11-14
> **Response, Clarification, and Completed Revisions (2/4)**
>
> **Review**
>
> > The paper doesn’t discuss the related work to a sufficient depth. In particular, other methods of evaluating counterfactual reasoning and whether there have been attempts at decomposing this process into smaller steps in evaluation is not discussed.
>
> ---
>
> **Response**
>
> We appreciate the reviewer’s suggestion to further highlight our contribution within the broader literature on counterfactual reasoning evaluation. To our knowledge, while existing literature offers important efforts in evaluating counterfactual reasoning in LLMs, most prior works assess it either (i) through isolated, static tasks such as contrastive counterfactuals (e.g., “What would happen if X didn’t occur?”) [1-5] or (ii) in tightly bounded settings such as causal QA or math word problems [6-8]. However, these evaluations rarely aim to explicitly understand  counterfactual reasoning into interpretable, structural chains of sub-tasks (e.g., variable identification, graph construction, intervention recognition, and counterfactual outcome estimation).
>
> Our major contribution is to operationalize Pearl’s structural counterfactual framework [9] into four modular and interpretable sub-tasks, with each task mapped to a step in the counterfactual reasoning pipeline. This decomposition offers an isolated view of failure modes in LLM reasoning. For instance, we demonstrate that even when an LLM correctly identifies variables (Task 1), it may fail in downstream causal graph construction (Task 2) or hallucinate interventions (Task 3). This level of task-aligned modularity is novel and important for understanding LLMs’ reasoning brittleness beyond flat accuracy metrics.
>
> ---
>
> **Completed Revision** (in Updated Manuscript)
>
> To address reviewer’s comments, we have updated our “Related Work” section (with highlighted color) accordingly to reflect this literature more clearly, as shown below.
>
> > Evaluating counterfactual reasoning in LLMs has garnered growing attention. However, most prior works assess counterfactual through end-to-end evaluations such as contrastive counterfactuals (e.g., ``What would happen if X didn’t occur?'') [1,2,4,5]. Other studies such as MalAlgoQA [6] introduce the concept of algorithms to assess LLMs’ ability to reason about flawed hypothetical paths. Their setup focuses on identifying distractor rationales in multiple-choice formats, revealing LLM struggles in understanding student misconceptions Other efforts like DICE [7] and CausalProbe [8] create diagnostic benchmarks to evaluate causal sensitivity or counterfactual faithfulness in static question-answering formats, without decomposing the reasoning process into interpretable modules . Similarly, synthetic datasets have been used to analyze RNN inductive biases in agreement prediction [3] but do not extend to structured counterfactual inference tasks. Compared to these works, our study introduces a modular evaluation aligned with Pearl’s structural causal model [9]. We decompose counterfactual reasoning into four interconnected sub-tasks to open up fine-grained attribution of model failures and provide a more diagnostic and interpretable assessment of LLM reasoning capabilities.
>
>
> **Reference**
>
> [1] CRASS: A Novel Data Set and Benchmark to Test Counterfactual Reasoning of Large Language Models
>
> [2] CLOMO: Counterfactual Logical Modification with Large Language Models
>
> [3] Studying the Inductive Biases of RNNs with Synthetic Variations of Natural Languages
>
> [4] What If the TV Was Off? Examining Counterfactual Reasoning Abilities of Multi-modal Language Models
>
> [5] COCO-Counterfactuals: Automatically Constructed Counterfactual Examples for Image-Text Pairs
>
> [6] MalAlgoQA: Pedagogical Evaluation of Counterfactual Reasoning in Large Language Models and Implications for AI in Education
>
> [7] DICE: A Framework for Dimensional and Contextual Evaluation of Language Models
>
> [8] Unveiling causal reasoning in large language models: Reality or mirage?
>
> [9] Judea Pearl. Causal inference in statistics: An overview

---

> ### Author Response · Authors · 2025-11-14
> **Response, Clarification, and Completed Revisions (3/4)**
>
> **Review**
>
> > The four sub-tasks that are identified, while generally sound, are arbitrary. Their choice is not motivated
>
> ---
>
> **Response**
>
> We appreciate the reviewer’s observation and would like to clarify the motivation behind our decompositional design.
> The four sub-tasks. (1) Causal Variable Identification, (2) Causal Graph Construction, (3) Intervention Identification, and (4) Outcome Reasoning, are not arbitrarily chosen. Rather, they are directly grounded in the structural formulation of counterfactual reasoning in the causal inference literature, especially the structural causal model (SCM) formalism proposed by Pearl [1]:
> - (i) inferring latent variables from observations,
> - (ii) modifying the model via intervention,
> - (iii) and computing the updated outcome.
>
> In the beginning of Section 3.2 in our manuscript, we highlight that our decomposition refines this chain into modularized evaluation tasks that map to the capabilities expected from LLMs engaging in counterfactual reasoning.
>
> In detail:
> - Task I (Causal Variable Identification) aligns with latent variable inference, in which the LLM must infer the presence and role of the core variables (Exposure, Covariate, Mediator, Outcome) from a given factual context.
> - Task II (Causal Graph Construction) corresponds to constructing the underlying structural model, formalized as a direct acrylic graph, which is required to support interventional semantics.
> - Task III (Intervention Identification) reflects the modification step, in which a hypothetical manipulation is proposed and the LLM must correctly interpret which variable is intervened upon.
> - Task IV (Outcome Reasoning) reflects the outcome computation under intervention, where the model simulates outcomes after the intervention by propagating effects along the updated causal paths.
>
> ---
>
> **Completed Revision** (in Updated Manuscript)
>
> To address the reviewer's comments, we have updated our “Introduction” section (with highlighted color) accordingly to further reflect our motivation, the added statements are shown below.
>
> > We are motivated by the structural formulation of counterfactual reasoning under the Structural Causal Model (SCM) formalism [1], in which counterfactual reasoning must proceed through a sequence of regularized steps—rather than open-ended inference—including inferring latent variables from observations, modifying the SCM via intervention, and computing the updated outcome. Accordingly, we outline a Decomposition Strategy…
>
> **Reference**
>
> [1] Judea Pearl. Causal inference in statistics: An overview

---

> ### Author Response · Authors · 2025-11-14
> **Response, Clarification, and Completed Revisions (4/4)**
>
> **Review**
>
> > While evaluations in Section 4.1 are interesting, they are not exactly actionable. They help identify which sub-tasks are the weak points to be improved (which arguably all a decomposition is supposed to do) and the paper suggests improvement for these weak points. However, the suggestions take the decomposition as a given. When the goal is to answer a question end-to-end, starting from the given of Task 1 and providing the outputs of Task 4, the paper does not explore the impact of going through Tasks 1, 2, 3, and 4 as a pipeline rather than direct queries. Moreover, it does not explore whether the task specific improvements in Section 4.2 lead to a comparable end-to-end gain as well.
>
> ---
>
> **Response and Completed Revision** (in Updated Manuscript)
>
> We thank the reviewer for these constructive suggestions. Regarding the comment that our evaluations are “not exactly actionable,” we would like to clarify that Section 4.1 serves as a diagnostic step to identify performance bottlenecks, which in turn motivate the targeted improvement strategies introduced in Section 4.2.
>
> We also appreciate the reviewer’s interest in end-to-end evaluation, specifically, “when the goal is to answer a question end-to-end, starting from the given of Task 1 and providing the outputs of Task 4.” In response, we have conducted additional experiments to assess how end-to-end performance changes when incorporating the improvements discussed in Section 4.2.
>
> We have included a new subsection (Section 4.2.3) detailing these end-to-end evaluations. The accompanying table presents the results, in which we integrate the proposed improvements for explicit variable identification (Section 4.2.1), implicit variable reasoning (Section 4.2.2), and their combined application to assess overall gains in counterfactual reasoning performance. The performance change is compared with Table 3 (where the results of Y' demonstrate the final counterfactual reasoning outcomes)
>
> Below are the evaluation results (with a same Table in added Section 4.2.3).
>
>
>
> | **Dataset**     | **GPT-5**    | **Qwen3** | **Llama-4S** | **Gemini-2.5** | **GPT-5**  | **Qwen3** | **Llama-4S** | **Gemini-2.5**| **GPT-5**       | **Qwen3** | **Llama-4S** | **Gemini-2.5** |
> | ---  | --- | ---  | ---  | ---  | ---  | --- | ---  | ---  | ---  | ---  | ---  | ---  |
> |   | **Improve Explicit Variable** |    |   |           | **Improve Implicit Variable** |     |         |           | **Improve Both** |         |         |           |
> | **CRASS**       | +1.8                          | +2.3    | +3.5    | +2.0      | +6.2                          | +7.4    | +7.9    | +6.8      | +9.0             | +10.1   | +10.5   | +9.3      |
> | **CLOMO**       | +2.1                          | +3.1    | +4.2    | +2.6      | +5.7                          | +6.3    | +6.1    | +6.5      | +9.2             | +10.4   | +11.0   | +9.8      |
> | **COCO**        | +1.2                          | +1.5    | +1.9    | +1.4      | +4.1                          | +5.1    | +5.9    | +5.1      | +5.8             | +6.4    | +6.7    | +6.2      |
> | **Open-Critic** | +1.6                          | +2.3    | +2.7    | +2.0      | +3.3                          | +8.3    | +11.2    | +4.3    | +5.3             | +11.5    | +14.2    | +5.5      |
>
>
>
> We have two observations:
> - (i) In general, improving implicit variable reasoning (i.e., for mediators and outcomes) yields more substantial gains in end-to-end performance, as these variables directly influence the final counterfactual predictions (Task IV). In contrast, improvements in explicit variable identification (e.g., for X, Z, Y) primarily strengthen the early stages of the pipeline, offering moderate but necessary support. - (ii) Notably, the combined strategy achieves the highest overall improvement, although the gains are not strictly additive. This is due to accumulative reasoning errors across the decompositional steps, where inaccuracies in earlier predictions may propagate and compound through the pipeline.
>
> These findings highlight both the opportunities and challenges of modular improvements.
>
> We have added the paragraphs into Section 4.2.3, with color texts highlighting our completed revisions.
>
> ---
> ---
>
>
> **Thank you very much**
>
> We would like to thank the reviewer again for their time and thoughtful feedback. We hope that the above clarifications and revisions address the reviewer’s concerns and contribute to improving the overall quality of the paper. We look forward to any further feedback you may have.

---

> ### Author Response · Authors · 2025-11-26
>
> Dear Reviewer,
>
> We would like to thank you again for your time and thoughtful feedback on our submission.
>
> We believe that the overall comments and questions primarily relate to the clarity of methodological details and auxiliary evaluations with discussions. **These points have been addressed through our point-by-point responses, along with the corresponding revisions in the updated manuscript.**
>
> We would very much welcome any further questions, concerns, or suggestions for revision, as we believe such feedback would help further re-evaluate the quality and clarity of our work.
>
> Thank you for your time and consideration.

---

> ### Author Response · Authors · 2025-11-27
>
> Dear Reviewers,
>
> We would like to thank you again for your time and thoughtful feedback on our submission.
>
> We believe that the overall comments and questions primarily relate to the clarity of benchmark details as well as auxiliary evaluations and discussions. **These points have been addressed through our point-by-point responses, along with the corresponding revisions (and added results) in the updated manuscript.**
>
> We would very much welcome for any further questions, concerns, or suggestions for revision, as we believe such feedback would help further re-evaluate the quality and clarity of our work.
>
> Thank you for your time and potential re-consideration.

---

### Meta-Review · Area_Chair_bmRJ · 2026-01-08

**Summary:**

The paper studies LLMs behavior on counterfactual questions ("What if ... ?" style questions) from the causality perspective. The paper introduces a benchmark of 11 datasets spanning math, coding, general QA, and text-image QA and tracks the performance along four tasks motivated by the causal inference literature.

Reviewers are generally positive but highlight several concerns:
* Benchmark design (aJkt, fFFZ) - motivation behind the selected four decomposition tasks and four causal variables
* Mapping benchmark performance to real-world use-cases and actionable insights (aJkt, BrYZ)
* Surface-level explanations of the observed phenomena (BrYZ)

Overall, I think the authors did a good job addressing those and elaborating on the motivation and related work, benchmark construction, adding a new experiment, so I would recommend an accept.

**Reviewer Concerns:**

* Benchmark design - the authors provided compelling evidence of grounding the benchmark design in the causal inference literature and clarified annotation and labeling scheme
* Real-world use-cases and actionable insights - the authors added results of the end-to-end pipeline tracking all four tasks and elaborated on the practical applications of measuring counterfactual reasoning capabilities in off-the-shelf LLMs
* Surface-level explanations of the results - this one remains open, the authors' explanation is vague and rather hand-wavy, I am not fully satisfied with the response.

That said, most of the concerns seem to be resolved.

**Reviewer Scores:**

* fFFZ 6 -> 8
* Ndd6, BrYZ kept 6, would probably have kept a positive score
* aJkt: I think their concerns have been addressed to a larger extent, so they would probably have increased to a positive score

---

### Decision · Program_Chairs · 2026-01-26

Accept (Poster)